# Structure-based discovery of hydrocarbon-stapled paxillin peptides that block FAK scaffolding in cancer

Lauren Reyes[1,10], Lena Naser[1,10], Warren S. Weiner[1,10], Darren Thifault[2,3,10], Erik Stahl[1], Liam McCreary[1], Rohini Nott[1], Colton Quick[4], Alex Buchberger[2,5], Carlos Alvarado[1], Andrew Rivera[1,6], Joseph A. Miller[1], Ruchi Khatiwala[1], Brian R. Cherry[7], Ronald Nelson[1,4], Jose M. Martin-Garcia[8], Nicholas Stephanopoulos[2,5], Raimund Fromme[2,3], Petra Fromme[2,3], William Cance[1] & Timothy Marlowe[1,4,6,9] ✉

The focal adhesion kinase (FAK) scaffold provides FAK-targeted cancer therapeutics with greater efficacy and specificity than traditional kinase inhibitors. The FAK scaffold function largely involves the interaction between FAK's focal adhesion targeting (FAT) domain and paxillin, ultimately regulating many hallmarks of cancer. We report the design of paxillin LD-motif mimetics that successfully inhibit the FAT-paxillin interaction. Chemical and biochemical screening identifies stapled peptide 1907, a high affinity binder of the FAT four-helix bundle with ~100-fold greater binding affinity than the native LD2-sequence. The X-ray co-crystal structure of the FAT-1907 complex is solved. Myristoylated 1907-analog, peptide 2012, delocalizes FAK from focal adhesions, induces cancer cell apoptosis, reduces in vitro viability and invasion, and decreases tumor burden in B16F10 melanoma female mice. Enzymatic FAK inhibition produces no comparable effects. Herein, we describe a biologically potent therapeutic strategy to target the FAK-paxillin complex, a previously deemed undruggable protein-protein interaction.

Focal adhesion kinase (FAK) is a dual protein tyrosine kinase-scaffolding protein that is a critical regulator of focal adhesion dynamics and cellular interactions between the cytoskeleton and the extracellular matrix. FAK is overexpressed in a large number of human cancers (50–80% of all solid tumors[1]) and modulates multiple hallmarks of cancer, including invasion, metastasis, resistance to apoptosis, proliferation, glycolysis, angiogenesis, and immune inactivation[2–5]. Due to its ubiquitous overexpression in human cancer and low expression in normal tissue, FAK has been a highly investigated cancer drug target for the past 20 years. In addition, FAK has been proposed as a target for fibrosis, due to its upregulation in fibrotic diseases[6,7] and role in modulating myofibroblast transformation, extracellular matrix (ECM) remodeling, and fibroblast migration[8,9].

Inhibitors of FAK that target the ATP-binding pocket of the kinase domain have been largely unsuccessful[10,11], displaying micromolar

[1]University of Arizona College of Medicine – Phoenix, Phoenix, AZ 85004, USA. [2]School of Molecular Sciences, Arizona State University, Tempe, AZ 85284, USA. [3]Center for Applied Structural Discovery, Arizona State University, Tempe, AZ 85287, USA. [4]FAKnostics, LLC, Phoenix, AZ 85004, USA. [5]Center for Molecular Design and Biomimetics, the Biodesign Institute, Arizona State University, Tempe, AZ 85287, USA. [6]Molecular Discovery Core, University of Arizona College of Medicine – Phoenix, Phoenix, AZ 85004, USA. [7]The Magnetic Resonance Research Center, Arizona State University, Tempe, AZ 85287, USA. [8]Crystallography & Structural Biology, Institute of Physical Chemistry Blas Cabrera, Madrid 28006, Spain. [9]Pharmacology and Toxicology, University of Arizona College of Pharmacy – Phoenix, 650 E. Van Buren St, Phoenix, AZ 85004, USA. [10]These authors contributed equally: Lauren Reyes, Lena Naser, Warren S. Weiner, Darren Thifault. ✉e-mail: tmarlowe@arizona.edu

anti-proliferative effects and no apoptotic effects in cancer cells despite reported nanomolar biochemical potencies for the FAK enzyme[12,13]. Furthermore, the efficacy of FAK kinase domain inhibitors in Phase I/II oncology clinical trials as monotherapy has been limited[10,11,14]. The ineffectiveness of FAK kinase inhibitors has been attributed to FAK's fundamental role as a scaffolding protein[15,16]. Through interactions with various protein binding partners, the FAK scaffold is believed to be the main modulator of the pro-survival processes of focal adhesions, including regulation of anoikis[17], apoptosis[17–19], focal adhesion assembly[20], and integration of oncogenic signaling complexes[21]. Specifically, the focal adhesion targeting (FAT) domain of FAK is a key scaffolding region responsible for FAK localization to focal adhesions, cell adhesion, and cancer cell survival[19,22,23]. Numerous studies utilizing adenoviral transduction of FAK-related non-kinase (FRNK), an endogenous dominant-negative inhibitor of the FAK FAT domain, showed displacement of FAK from focal adhesions, leading to increased cell rounding, detachment, and apoptosis[18,19]. Thus, the FAK FAT domain is an integral mediator of FAK function within the focal adhesion complex. Despite these promising data, a potent and structurally validated direct inhibitor of the FAK scaffold has yet to be identified.

The interaction between the FAT domain and paxillin is critical for mediating FAK's scaffolding activity and subsequent oncogenic survival processes. Paxillin's leucine-aspartic acid (LD) alpha-helical motifs, LD2 and LD4, are specifically involved in FAT binding[24]. The FAT domain contains a four-helical bundle, in which paxillin LD2 and LD4 interact at two separate hydrophobic patches between helices 1–4 and 2–3 on the FAT domain. The FAT helix 2–3 site is primarily responsible for full-length paxillin binding[25], however, disruption of both helix sites is required for maximum biological effect[25–27]. Mutations at the helix 2–3 and helix 1–4 FAT-paxillin binding sites inhibit FAK localization, FAK and paxillin phosphorylation, focal adhesion turnover, cell adhesion, migration, and invasion[25,26,28]. Therefore, disrupting the FAT-paxillin protein–protein interaction (PPI) is a desirable strategy to inhibit the FAK pathway in cancer.

Protein–protein interactions have traditionally been viewed as chemically intractable targets because their critical interaction surfaces typically lack the compact, deep, hydrophobic involutions that enable potent, selective binding[29]. Moreover, many PPI targets are intracellular, putting them beyond the reach of protein biologics (monoclonal antibodies, etc.). While PPIs have previously been considered undruggable targets, the all-hydrocarbon-stapled alpha-helical peptide has emerged as a promising drug class to target such interactions. Hydrocarbon-stapled peptides are covalently constrained alpha-helical peptidomimetics, synthesized by substitution of two strategically spaced residues with terminal olefin-containing unnatural amino acids, followed by ring-closing olefin metathesis (RCM) to cyclize the peptide. Three staple spacings ($i,i+3$; $i,i+4$; and $i,i+7$) and corresponding olefinic amino acid pairs are utilized to occupy coplanar positions on the alpha-helix, producing high yields after RCM[30]. Hydrocarbon-stapled peptides attain similar cell permeability to small molecules, while exhibiting the broad target recognition capabilities typical in protein therapeutics[31]. Furthermore, while canonical helical peptides lose their conformation in free solution, stapled peptides are locked into their secondary structure and generally retain their helicity even in harsh or destabilizing conditions[32]. Hydrocarbon stapling also enhances the pharmacological performance of peptide drugs by dramatically improving target affinity, resistance to proteolysis, cell permeability, and in vivo pharmacokinetics[33,34].

Here, we report the design of a promising class of FAK scaffold inhibitors, all-hydrocarbon-stapled peptides based on the paxillin LD2 alpha-helical motif, that target the FAT-paxillin interaction. We report the X-ray co-crystal structure of a FAK scaffold inhibitor in a complex with the FAK FAT domain (PDB 6PW8). Based on molecular modeling of the FAT-paxillin interaction, we synthesized a collection of stapled peptides with varied staple positions and spacings, and derivative analogs with rational residue substitutions. Using surface plasmon resonance (SPR) binding and fluorescence polarization (FP) competition assays, we obtained a base compound, 1907, with enhanced binding capabilities compared to native LD2. Biophysical characterization, including heteronuclear single quantum coherence-nuclear magnetic resonance (HSQC-NMR) experiments and co-crystallization of 1907 with the FAT domain, revealed improved binding interactions of peptide 1907 in comparison to the FAK FAT-paxillin LD2 (FAT-LD2) co-crystal. We further modified 1907 by N-myristoylation to identify an optimized lead peptide (2012) with improved cell permeability, drug metabolism, pharmacokinetic (DMPK) properties, and in vitro efficacy. 2012 exemplified the expected effects of delocalizing FAK from focal adhesions in cancer cells: reduced viability, induction of apoptosis, and reduced invasion. Furthermore, 2012 displayed excellent in vivo pharmacokinetics, and greatly reduced tumor burden in the B16F10 melanoma mouse model. In summary, we report a feasible strategy to target elusive PPIs and its specific application to the design of effective therapeutics targeting the FAK-paxillin interaction, central to multiple hallmarks of cancer.

## Results

### Stapled peptide design and structure–activity relationships

To identify inhibitors of the FAT domain of FAK, we employed the hydrocarbon-based stapling approach in the design of peptide analogs with drug-like features such as increased stability, proteolytic resistance, binding affinity, cell permeability, and DMPK properties. The primary template (peptide 1; herein referred to as LD2) was based off the paxillin LD2 sequence ([141]NLSELDRLLLELN[153]), which is a known FAK FAT binder[21,22,30] and has been co-crystallized with the FAK FAT domain (Fig. 1a). Several key features of this sequence were observed and exploited in the peptide design (Fig. 1b, PDB 1OW8), including key hydrophobic residues (L145, L148, L149, and L152) and polar/charged residues (D146, E151, and N153) that were responsible for providing structural specificity to the FAT-paxillin interaction. Initial identification of structure–activity relationship (SAR) efforts focused on staple scanning to identify the optimal position and spacing of the hydrocarbon stapling moiety. We varied the stapling motif ($i,i+3$; $i,i+4$; $i,i+7$) and position on the LD2 sequence. Analogs were designed and synthesized with the goal of stabilizing helical structure and improving favorable intermolecular interactions (Supplementary Figs. 1, 2 and Supplementary Table 1). A peptide (2; herein referred to as 1907) with an $i,i+7$ staple in place of residues S143 and L150, located at the helical face opposite the hydrophobic interface, exhibited significantly better activity compared to LD2 in SPR binding and FP competition assays (Table 1 and Fig. 2). LD2 fit a single-site $K_D$ of $82.6 \pm 7.7\,\mu M$ by SPR and showed a $K_i$ of $70.6 \pm 31.8\,\mu M$ by FP. In contrast, 1907 demonstrated two binding sites ($K_{D1} = 0.8 \pm 0.6\,\mu M$, $K_{D2} = 8.7 \pm 3.3\,\mu M$), with a 100-fold increased affinity over LD2, and a tenfold improved $K_i$ of $5.8 \pm 4.6\,\mu M$ by FP (Table 1 and Fig. 2a, b). Importantly, peptide analogs with stapling moieties present at the hydrophobic interface (1914, 1919, 1921, 2015, 2017) had limited improvement or even loss of activity compared to LD2, suggesting that substitution of these interfacial residues disrupted key hydrophobic interactions (Table 1). Furthermore, 1905, which contains an $i,i+7$ staple in place of E144 and E151, both of which are important for forming electrostatic interactions, performed poorly in SPR and FP assays (Supplementary Figs. 3, 4). Overall, these data identified peptide 1907 as a base compound with potent FAT binding and established that a hydrocarbon bridge between amino acid positions 143 and 150 provided an increased affinity and improved inhibitory activity of 1907 over the native LD2 sequence.

Subsequent SAR studies were focused on the optimization of 1907 and the identification of residues that were critical for peptide activity (Table 1). To probe additional protein-peptide contacts by lengthening the peptide, we synthesized an analog with an extra helical turn of the

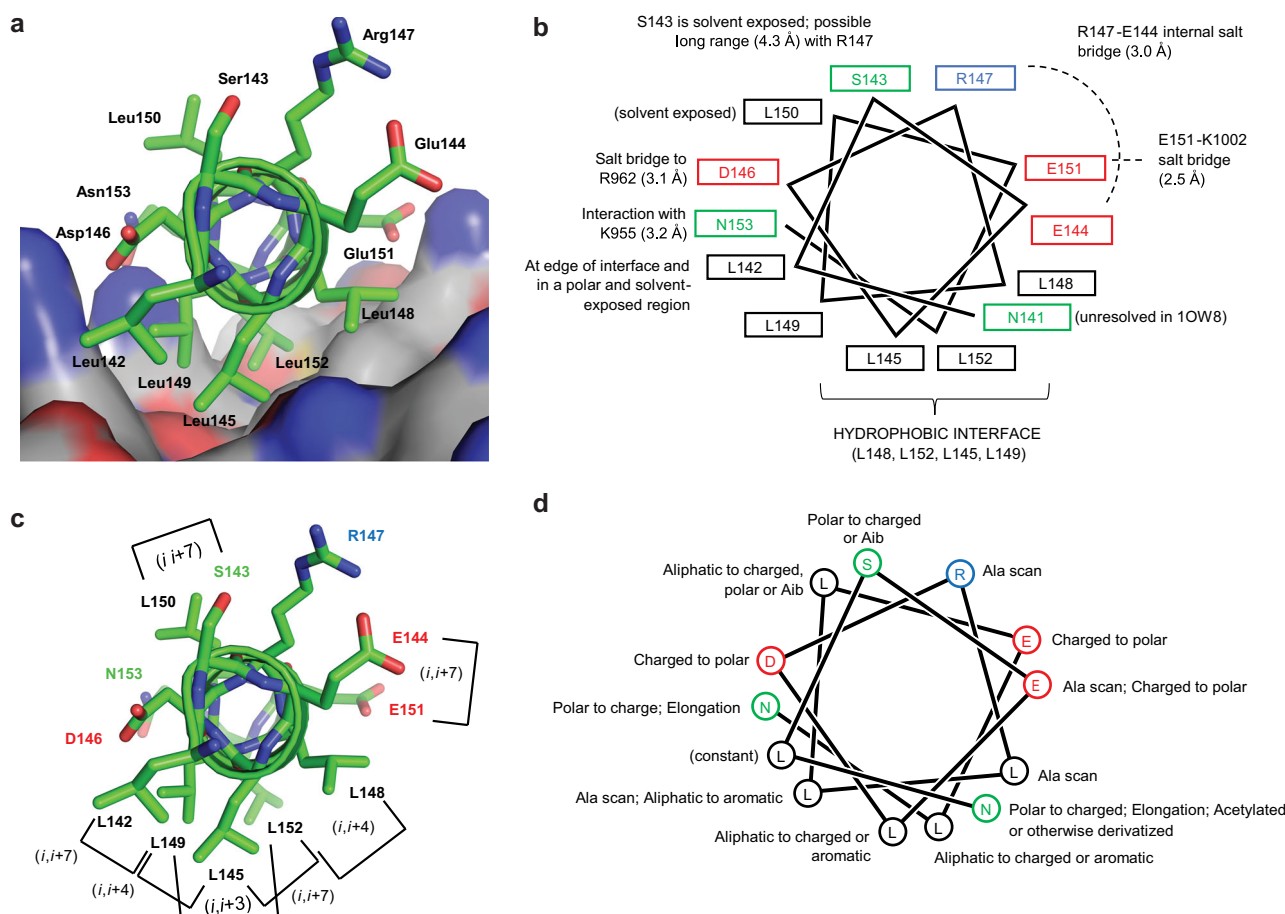

**Fig. 1 | Structure-based design of cyclic alpha-helical peptides based on the native paxillin LD2 sequence. a** The alpha-helical LD2 motif ([141]NLSELDRLLLELN[153]) from X-ray crystal structure, PDB 1OW8[24]. **b** Structural analysis summary of LD2-native FAT intra- and intermolecular interactions. **c** Stapling strategies. **d** Residue modifications, excluding staples, as used in peptide design strategies.

native sequence at both the N- and C-terminus (1920). Peptide 1920 demonstrated similar two-site binding affinity ($K_{D1} = 0.9 \pm 0.5\,\mu M$, $K_{D2} = 9.8 \pm 6.9\,\mu M$) but less competitive inhibition ($K_i = 25.0 \pm 1.8\,\mu M$) than 1907, indicating that the additional residues do not significantly improve peptide activity. Analysis of a truncated peptide (2011) indicated that residues N- and C-terminal to the $i,i+7$ staple in 1907 were critical for activity, which is consistent with the co-crystal model of the FAT-LD2 interaction. Further, SAR experiments focused on residue-by-residue optimization of 1907 to identify residues that were critical to activity. Keeping the hydrophobic interface constant, we modulated the flanking polar residues (N141, E144, R147, E151, and N153) to increase logP for enhanced cell permeability and to minimize dependence on charge-charge interactions. To this end, we synthesized peptides 1910, 2009, 2010, and 1913 with substitutions at these positions (Table 1). Intriguingly, SPR indicated that these peptides displayed lower binding affinities and had reduced competitive inhibition of FAT-LD2 binding.

Substitution of key leucine residues (L145, L149, and L152) in the hydrophobic interface with tryptophan (1933 and 2007), significantly reduced peptide activity, indicating that these leucine residues are required for interaction with the hydrophobic pockets of FAT. Synthesis of a peptide containing aminoisobutyric acid (Aib) as an alternative alpha-helical stabilizing strategy (1912) did not improve activity relative to stapled 1907. The cyclization of the hydrocarbon-stapled peptide by RCM was also important, as 2013, the unstapled version of 1907 with no other modifications, did not bind FAT nor did it inhibit FAT-LD2 binding to the same level as 1907 (Table 1 and Fig. 2c). These

data highlighted the benefit of engineering sequence specificity and hydrocarbon stapling to increase the activity of peptide 1907, and also informed the design of our in vitro negative control peptide (2014) with L145E and L152E substitutions, which did not bind FAT in SPR or FP experiments (Table 1 and Fig. 2d). In all, ten stapled peptides demonstrated better binding to the FAT domain than the native paxillin LD2 sequence in SPR assays. Furthermore, nine stapled peptides were able to disrupt FAT-paxillin interactions at a better $K_i$ compared to LD2 in FP analysis. One peptide, 1907, demonstrated superior FAT activity compared to the others and was pursued for additional optimization and characterization.

## NMR validation of 1907 interactions and circular dichroism
To validate the binding mode of 1907 and map its interactions with the FAT domain, we performed heteronuclear single quantum coherence-nuclear magnetic resonance (HSQC-NMR) chemical shift perturbation (CSP) studies with [15]N labeled recombinant FAT protein and titrating amounts of 1907 from 100 nM to 200 μM (Fig. 3a). Given 1907's higher affinity for FAT (SPR: $K_{D1} = 0.8\,\mu M$, $K_{D2} = 8.7\,\mu M$) compared to native LD2 (82.6 μM), we expected this interaction to adopt slow-exchange NMR features, where ligand binding induces the disappearance and appearance of specific chemical peaks correlated with unbound and bound conformations, respectively. Indeed, after adding 1907, the FAT HSQC spectra showed several correlation peaks that decreased in a concentration-dependent manner. Plotting and mapping the changes in CSPs when bound with 6 μM 1907 compared to DMSO control demonstrated that 1907 was able to induce CSP differences primarily

**Table 1 | Peptide structure–activity relationships (SAR) identify key requirements for biological activity and facilitate peptide optimization**

| ID | Class | N-mod | 141 | 142 | 143 | 144 | 145 | 146 | 147 | 148 | 149 | 150 | 151 | 152 | 153 | C-mod | SPR $K_D \pm$ SD (µM) | FP $K_i \pm$ SD (µM) |
|---|---|---|---|---|---|---|---|---|---|---|---|---|---|---|---|---|---|---|
| **1** (UA-1967) | LD2 | Ac- | N | L | S | E | L | D | R | L | L | L | E | L | N | -NH₂ | 82.6 ± 7.7 | 70.6 ± 31.8 |
| **2** (UA-1907) | Staple Scan | Ac- | N | L | $R_8$ | E | L | D | R | L | L | $S_5$ | E | L | N | -NH₂ | SITE1 0.8 ± 0.6 / SITE2 8.7 ± 3.3 | 5.8 ± 4.6 |
| **3** (UA-1914) | | Ac- | N | L | S | E | $R_8$ | D | R | L | L | L | E | $S_5$ | N | -NH₂ | 120 ± 67 | > 1250 |
| **4** (UA-1919) | | Ac- | N | L | S | E | $S_5$ | D | R | L | $S_5$ | L | E | L | N | -NH₂ | 80.6 ± 24.2 | 38.6 ± 2.3 |
| **5** (UA-1921) | | Ac- | N | L | S | E | L | D | R | L | $R_5$ | L | E | $S_5$ | N | -NH₂ | > 200 | 492 ± 96 |
| **6** (UA-2015) | | Ac- | N | L | S | E | L | D | R | $S_5$ | L | L | E | $S_5$ | N | -NH₂ | 30.6 ± 17.8 | 114 ± 11.3 |
| **7** (UA-2017) | | Ac- | N | $R_8$ | S | E | L | D | R | L | $S_5$ | L | E | L | N | -NH₂ | 27.9 ± 8.1 | 80.6 ± 9.4 |
| **8** (UA-1905) | | Ac- | N | L | S | $R_8$ | L | D | R | L | L | L | $S_5$ | L | N | -NH₂ | 145 ± 64.7 | > 1250 |
| **9** (UA-1920) | SP Optimization | AcSLGS- | N | L | $R_8$ | E | L | D | R | L | L | $S_5$ | E | L | N | -AVQHNH₂ | SITE1 0.9 ± 0.5 / SITE2 9.8 ± 6.9 | 25.0 ± 0.77 |
| **10** (UA-2011) | | Ac- | | | $R_8$ | E | L | D | R | L | L | $S_5$ | | | | -NH₂ | > 200 | > 333 |
| **11** (UA-1910) | | Ac- | N | L | $R_8$ | Q | L | D | R | L | L | $S_5$ | Q | L | N | -NH₂ | 22.6 ± 4.4 | 46.8 ± 17.9 |
| **12** (UA-2009) | | Ac- | N | L | $R_8$ | A | L | D | R | L | L | $S_5$ | E | L | N | -NH₂ | 3.4 ± 3.1 | 26.7 ± 10.8 |
| **13** (UA-2010) | | Ac- | N | L | $R_8$ | A | L | D | A | L | L | $S_5$ | E | L | N | -NH₂ | 12.6 ± 0.7 | 68.5 ± 7.0 |
| **14** (UA-1913) | | Ac- | D | L | $R_8$ | E | L | D | R | L | L | $S_5$ | E | L | D | -NH₂ | 13.7 ± 2.9 | 21.8 ± 7.6 |
| **15** (UA-1933) | | Ac- | N | L | $R_8$ | E | W | D | R | L | L | $S_5$ | E | W | N | -NH₂ | 155 ± 100 | 281 ± 110 |
| **16** (UA-2007) | | Ac- | N | L | $R_8$ | E | L | D | R | L | W | $S_5$ | E | L | N | -NH₂ | 60.6 ± 26.3 | 190 ± 20.4 |
| **17** (UA-1912) | | Ac- | N | L | Aib | E | L | D | R | L | L | Aib | E | L | N | -NH₂ | 38.7 ± 4.0 | 28.1 ± 2.6 |
| **18** (UA-2013) | Control | Ac- | N | L | $R_8^*$ | E | L | D | R | L | L | $S_5^*$ | E | L | N | -NH₂ | 37.2 ± 2.8 | 20.3 ± 3.3 |
| **19** (UA-2014) | | Ac- | N | L | $R_8$ | E | E | D | R | L | L | $S_5$ | E | E | N | -NH₂ | > 200 | > 1250 |

Bolded amino acids indicate olefinic amino acids. Bolded peptide IDs indicate order of mention in the manuscript. UA codes refer to internal identification number utilized in compound database. Grayscale shaded amino acids indicate following properties: polar uncharged (light gray), negatively charged (gray), positively charged (dark gray). The reported $K_D$ values are the mean ± SD of 5 biological replicates ($n = 5$) for UA-1907; mean ± SD of 4 biological replicates ($n = 4$) for UA-1967, UA-1914, UA-1919, UA-2015, UA-2017, UA-1905, UA-1910, UA-1913, UA-2007, UA-1912, UA-2013, and UA-2014; mean ± SD of 3 biological replicates ($n = 3$) for UA-1921, UA-1920, UA-2011, UA-2009, and UA-1933; and mean ± SD of 2 biological replicates ($n = 2$) for UA-2010. All $K_i$ values reported are the mean ± SD of 3 biological replicates ($n = 3$). All source data for derived $K_D$ and $K_i$ values are provided as Source Data file.
*R8* (R)-2-(7-octenyl)alanine (unstapled), *S5* (S)-2-(4-pentenyl)alanine, *Aib* 2-aminoisobutyric acid.
*indicates unstapled peptide where ring-closing metathesis was not performed.

in helix 2–3, specifically residues: M953, K955, E956, L959, R962, T963, L990, L994, G995 (Fig. 3b, c). Also, significant CSP changes were observed in helix 1–4, specifically the residues S940, K941, and I942. These interactions demonstrated significant overlap with the known binding sites of native peptide paxillin LD2, indicating that 1907 binds to the same sites as LD2.

The residues in helix 2–3 (L959, K955) showed a more prominent decrease in peak area upon 1907 titration compared to those in helix 1–4 (I936, K1032), suggesting that helix 2–3 binding could be stronger than helix 1–4. To determine the NMR-derived $K_D$ estimations of 1907 at each FAT binding site, we integrated the peak areas of key residues important for LD2 binding at each site and plotted change in integration area as a function of 1907 concentration (Fig. 3d and Supplementary Fig. 6). Intriguingly, NMR analysis estimated a stronger $K_D$ of 1907 for helix 2–3 (L959 = 1.0 µM, K955 = 1.6 µM) than for helix 1–4 (I936 = 27.0 µM, K1032 = 25.7 µM). At helix 2–3 residues L959 and K955, binding saturation was observed as peaks completely disappeared at 10 µM of the peptide. However, at the helix 1–4 residues I936 and K1032, complete disappearance of peaks was not observed until 100 µM of 1907, leading to the conclusion that weaker binding was observed at the helix 1–4 site. At 200 µM of 1907, full saturation of the entire HSQC spectrum was not achieved, which is often not experimentally feasible but facilitates $K_D$ calculation[35]; however, residue-by-residue peak integration analysis showed further elucidation of the helix 1–4 site (Supplementary Fig. 7). Although slow-exchange NMR features do not evaluate $K_D$ as accurately as fast exchange[35], these $K_D$ estimations correlated well with two-site SPR binding data (Table 1). Next, circular dichroism (CD) was performed to compare the alpha-helicity of 1907 vs. LD2 or an unstapled version of 1907 (2013) (Supplementary Fig. 5). As expected, 1907 demonstrated increased helical

content (80%) vs. unstapled peptide 2013 (70%) and LD2 (71%). In summary, these studies confirmed that 1907 binds primarily to the helix 2–3 site on the FAT domain, with weaker binding to the helix 1–4 site. Further, cyclization of the peptide via hydrocarbon stapling increases the alpha-helicity relative to the native or unstapled peptide.

## Solving the 1907-FAT complex by X-ray co-crystallography

To better understand the molecular basis of the FAT-1907 interaction, we co-crystallized and solved the FAT-1907 structure at 1.95 Å resolution (PDB 6PW8, Fig. 4, Supplementary Table 2, and Supplementary Fig. 8). As previously described[24,36], the FAT domain folds into an antiparallel four-helical bundle, connected by short loop regions and a close-packed hydrophobic core (Fig. 4). Similar to LD2, 1907 binds to helices 2 (949–973) and 3 (978–1005) of the FAT domain through hydrophobic interactions with the FAT hydrophobic core and electrostatic forces/hydrogen bonding between side chains of the peptide and FAT. No electron density was detected at the helix 1–4 site of 1907, likely due to its lower affinity compared to the helix 2–3 site, as calculated by NMR and SPR.

The electron density map of 1907 reveals a well-resolved N-terminal region, stabilizing the protein-peptide interface, with a network of hydrogen bond interactions. Residue N141 of 1907 interacts with residues N991 and E984 of FAT (Fig. 4). Furthermore, water-mediated hydrogen bonds form additional interactions between N141 and L142 of 1907 with D969 of FAT. To identify any changes in hydrophobic interactions, an intermolecular contact analysis at the protein-peptide interface was performed on all residues within 4 Å for FAT-1907 and FAT-LD2 (Supplementary Tables 3, 4). Interestingly, we found that 1907 residues L142, L145, L148, and L149 form new hydrophobic interactions with FAT residues L965, L961, G995, and L959/L994, respectively, that

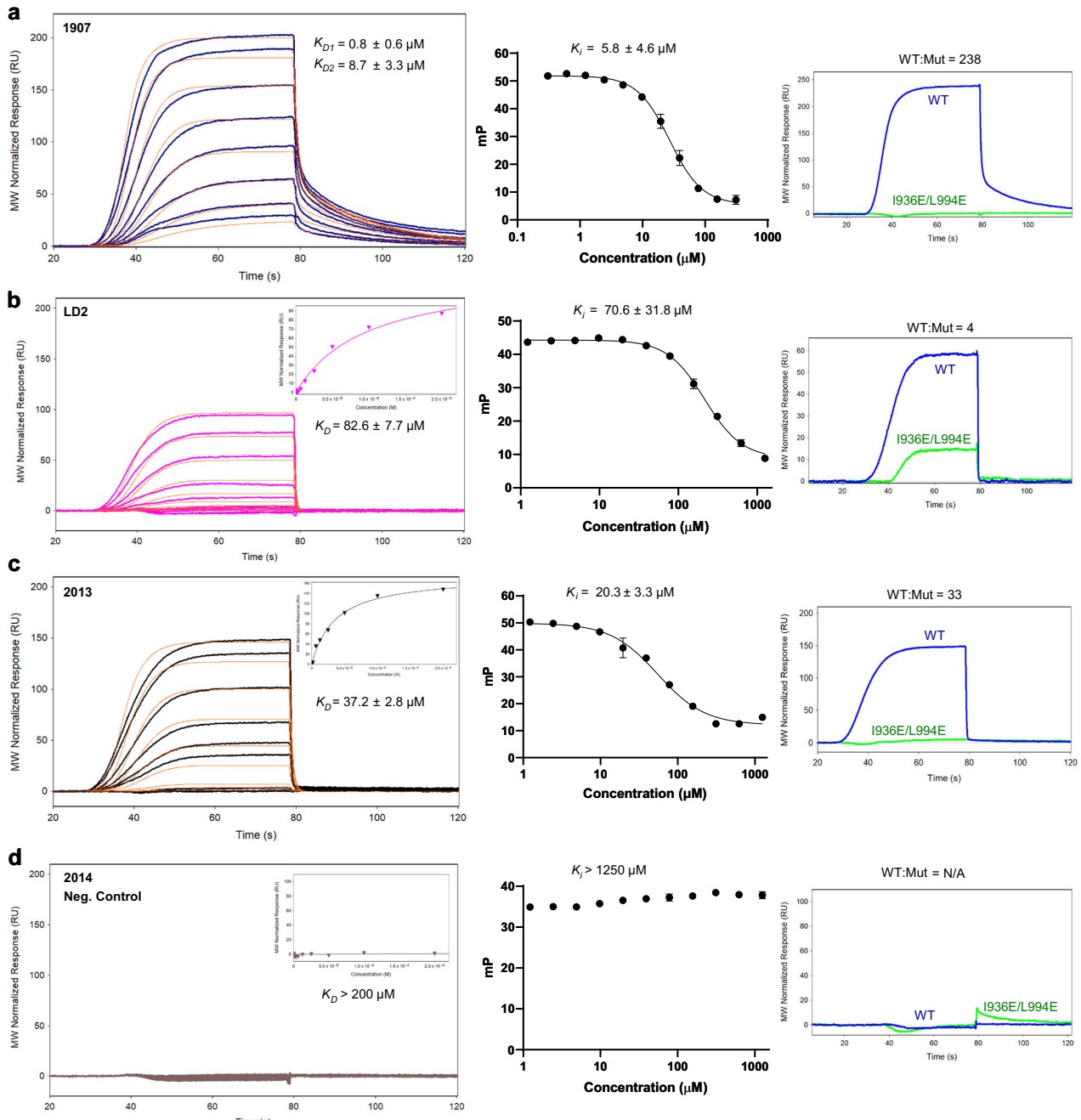

**Fig. 2 | Biophysical and biochemical characterization of synthetic peptides identifies peptide 1907 with improved binding affinity and inhibition of the FAK-paxillin complex.** SPR and FP data for **a** 1907, **b** LD2, **c** 2013, and **d** 2014. Left: SPR sensograms for peptide binding to immobilized FAK FAT, performed at peptide concentrations from 200 μM to 49 nM, with thick lines as response data and orange lines representing the kinetic fit. All peptides fit to a pseudo first-order one-site binding model to calculate the $K_D$ from kinetic data, other than peptide 1907, which fit to a two-site binding model. The equilibrium fit model is shown in the inset. For 1907, the $K_D$ reported is the mean ± SD of 5 biological replicates ($n = 5$).

For LD2, 2013, and 2014, the $K_D$ reported is the mean ± SD of 4 biological replicates ($n = 4$). Middle: FP competition assays using TAMRA-paxillin LD2 L10D and recombinant FAK FAT were performed at peptide concentrations from 1.25 mM to 1.22 μM, except for 1907, which was performed from 312.5 μM to 305.2 nM. The $K_i$ reported is the mean ± SD of 3 biological replicates ($n = 3$). Right: The selectivity between wild-type FAT and a FAT mutant (I936E/L994E) demonstrates the peptide's specific binding to the FAT domain. The source data are provided as a Source Data file.

were not present in the LD2 model. These additional intermolecular contacts may explain the increase in binding affinity and competition of peptide 1907.

Further structural comparison of our FAT-1907 structure to that of the FAT-LD2 complex shows that, overall, the arrangement of the FAT domain was mostly conserved (root-mean-square deviation: 0.603). The addition of the hydrocarbon staple resulted in minor conformational changes of the side chains of FAT residues K1002 and

R962 (Supplementary Fig. 9). Although the position of these side chains was altered, K1002 and R962 retained electrostatic interactions with residues E151 and D146, respectively, in 1907. These conformational changes allow water molecules to stabilize the complex by solvent-mediated hydrogen bonds between existing interactions at R962 and D146, as well as forming a new interaction between E151 with N999. An additional solvent-mediated hydrogen bond was also observed between K955 and L152 (Supplementary Fig. 9). Seven

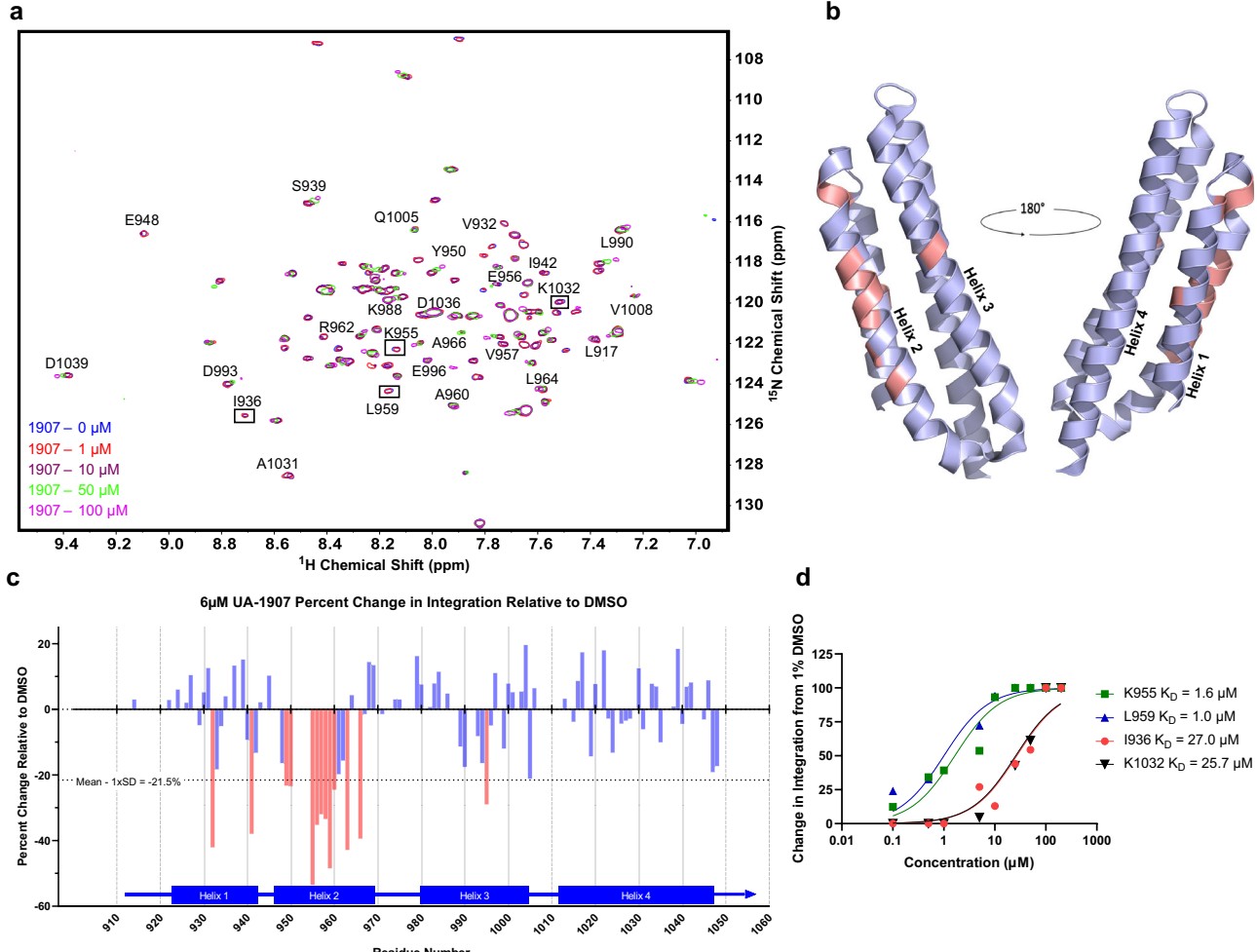

**Fig. 3 | Stapled peptide 1907 displays greater binding to FAT helix 2–3 site as demonstrated by HSQC-NMR studies. a** Overlay of 2D HSQC spectra acquired with varying concentrations of 1907 and 100 µM FAT. Two-dimensional signals represented in magenta, green, purple, red, and blue were acquired in the presence of 100, 50, 10, 1, and 0 µM of 1907 respectively. Residues that were previously shown to have notable chemical shift perturbations in the presence of LD2 and have notable differences in integration between high and low concentrations of 1907, are indicated. **b** Molecular model (PDB ID: 1OW8[24]) of the LD2 binding surfaces of FAT. Residues with a $^1$H-$^{15}$N HSQC peak volume decrease of more than 1 standard deviation upon incubation with 6 µM 1907 are highlighted in salmon. **c** Histogram of the residue-wise HSQC peak volume change, shown as percent relative to DMSO upon incubation of FAT with 6 µM 1907. Highlighted residues match those in panel (**b**). **d** Analysis and $K_D$ estimation of 1907 binding at residues I936, K955, L959, and K1032 from the peak integration studies shown in Supplementary Fig. S6. Source data are provided as a Source data file.

carbon atoms of the staple (C5-C11) were resolved at a density level 0.85$\sigma$ above the mean electron density, versus 0.25$\sigma$ for the remaining placement of carbon atoms (C1-C4) (Supplementary Fig. 10). Furthermore, the 2Fo-Fc electron density map and polder map confirm the presence of 1907 in the structure (Supplementary Fig. 11). As expected, the hydrocarbon-staple moiety did not interact with FAT but pointed outward from the complex interface. Additionally, we were able to model the double bond between carbons C7 and C8 in a trans-isomeric state. This X-ray structure reveals intermolecular interactions of the FAT-1907 complex that may explain the enhanced binding affinity of peptide 1907 and validate the proposed structural mechanism of action.

**Myristoylated 1907 analog (2012) exhibits drug-like features**
Myristoylation has been shown to increase the lipophilicity of peptides and facilitate cellular permeability[37]. To produce a stapled peptide lead with high cell permeability and favorable drug-like properties, peptide 1907 was derivatized with an N-terminal myristoyl group (2012, Fig. 5a and Supplementary Fig. 2). Negative control peptide 2014 was similarly modified (2020, Fig. 5a). To monitor cellular uptake, peptides were further modified by covalent tagging with Tetramethylrhodamine

(TAMRA) and tested in a plate-based fluorescence assay. The cellular uptake of TAMRA-tagged 2012 increased 6.5-fold in SK-MEL-147 melanoma cells and 12-fold in WM88 cells as compared to TAMRA-Alkyne (fluorophore only control, Fig. 5b). In comparison, the unmyristoylated parent peptide 1907 displayed 0.51-fold and 0.85-fold uptake in SK-MEL-147 and WM88 cells respectively compared to TAMRA-Alkyne alone. Peptide 2020 displayed cell permeability approximately threefold higher compared to the TAMRA-Alkyne control. Cell permeability and membrane localization of TAMRA-tagged 2012 (2036) was validated as well (Supplementary Fig. 12). Next, because peptide drugs are susceptible to proteolytic cleavage, we tested this series in a trypsin protease assay. Compared to the native LD2 peptide, which was rapidly degraded by trypsin protease ($t_{1/2}$ = 47 min), hydrocarbon-stapled peptides 1907 and 2012 displayed enhanced proteolytic resistance, with half-lives ($t_{1/2}$) of 28.4 h and >48 h, respectively (Fig. 5c).

**2012 disrupts FAK localization to focal adhesions**
After establishing lead peptide 2012 to be cell permeable and resistant to proteolytic cleavage, we began evaluating the ability of 2012 to act as a FAK FAT domain inhibitor and disrupt the cellular FAT-paxillin PPI.

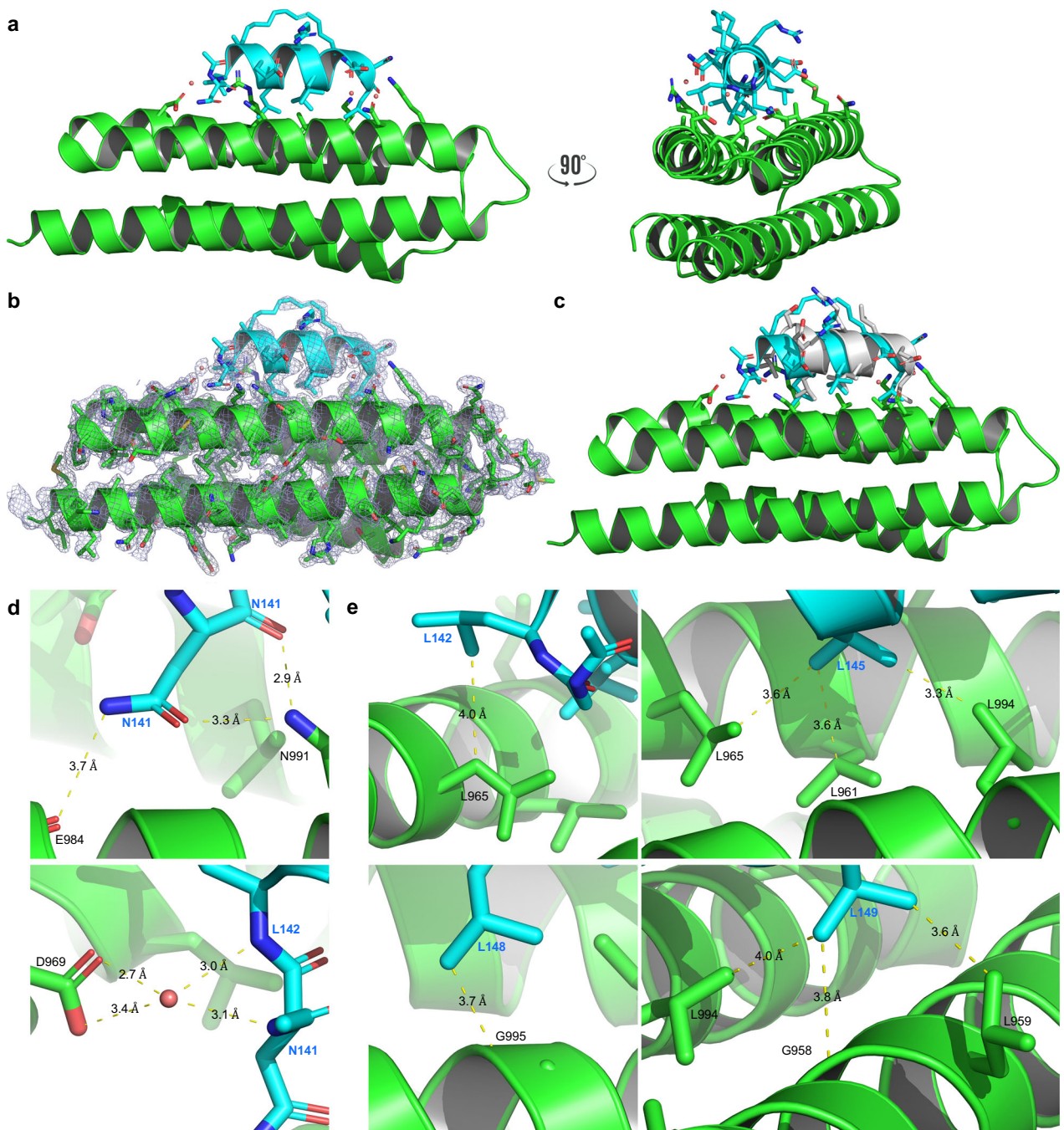

**Fig. 4 | X-ray co-crystal structure of the FAT-1907 complex reveals the binding mode to the high-affinity helix 2–3 site. a** Overall structure of co-crystallized 1907 peptide (cyan) bound to the FAK FAT domain (green); PDB 6PW8. Side chains of 1907 and the FAT binding site (K955, L959, L961, R962, L965, D969, N991, L994, N999, and K1002) are shown. **b** The 2Fo-Fc Electron density map at 0.85 sigma. **c** Superimposed model of LD2 (gray, PDB 1OW8[24]) with 1907 and the FAK FAT domain. **d**, top 1907 residue N141 forms direct hydrogen bonding interactions with FAT residues N991 and E984. **d**, bottom 1907 residues N141 and L142 additionally interact with FAT residue D969 through solvent (salmon sphere) mediated hydrogen bonding. **e** 1907 residues L142, L145, L148, and L149 form hydrophobic interactions with FAT residues L965, L961, G995, and L959/L994, respectively, not seen in previous models.

FAK localization to focal adhesions (FAs) is regulated through the interaction between the FAK FAT domain and key adapter protein, paxillin, whereas FAK kinase activity does not affect FAK localization[22,38]. We developed a confocal immunofluorescence assay to measure FAK and the FAT domain binding partner, paxillin, presence at FAs to assess whether lead peptide 2012 can effectively inhibit the FAT-paxillin PPI and delocalize FAK from FA complexes (Fig. 5d). Treatment with 1.5 µM 2012 in SK-MEL-147 cells reduced the number of FAK-containing FAs compared to the DMSO (vehicle) treated group and an accumulation of FAK in the cytoplasm was observed (Fig. 5d, e).

Paxillin containing FAs also decreased following treatment in 2012, although the reduction was slightly less compared to FAK-containing FAs (Fig. 5d, f). The amount of dual FAK and paxillin-containing FAs was also significantly reduced following treatment with our FAT domain inhibitor (Fig. 5d, g), collectively highlighting the ability of 2012 to inhibit the FAK-paxillin interaction and subsequent localization to FAs. The effects of FAT domain inhibition were then compared against the FAK kinase inhibitor, defactinib, which is the most clinically advanced FAK inhibitor in development and currently being investigated in Phase III clinical trials (NCT06072781). Interestingly, treatment with

1.5 μM defactinib showed a statistically significant increase in FAK, paxillin, and dual FAK and paxillin-containing FAs, highlighting a critical difference in targeting the FAT versus kinase domains of FAK (Fig. 5d–g). No statistically significant effects on FA composition were observed following treatment with control peptide 2020. Similar results were obtained in the cell line WM88 (Supplementary Figs. 14, 15).

The confocal immunofluorescence assay was subsequently utilized to evaluate the dose-dependent differences between FAK FAT inhibitor, 2012, and kinase domain inhibitor, defactinib, on FAK/paxillin FA localization. The decrease in FAK-containing FAs following treatment with 2012 was dose-dependent, yielding an IC$_{50}$ value of 0.80 μM (Fig. 5h and Supplementary Fig. 13), which was the same as the FAT binding affinity at the high-affinity site ($K_{D1}$ = 0.8 μM). Paxillin containing FAs also decreased in a dose-dependent manner following treatment with 2012, albeit with a higher IC$_{50}$ value of 4.41 μM (Fig. 5i and Supplementary Fig. 13). The amount of dual FAK-paxillin containing focal adhesions was also significantly reduced after treatment with 2012 in a dose-dependent manner with an IC$_{50}$ value of 1.26 μM (Fig. 5j). Additionally, disruption of the FAK-paxillin PPI by 2012 was further demonstrated in a mammalian two-hybrid assay in 293 T cells treated with various concentrations of 2012 (Supplementary Fig. 16). Treatment with defactinib showed an increase in FAK, paxillin, and dual FAK-paxillin containing focal adhesions from 0.1 to 3.1 μM, but at higher doses, defactinib was able to reduce the amount of FAK and paxillin containing FAs with an IC$_{50}$ value of 11.11 μM and 18.65 μM respectively (Fig. 5h, i and Supplementary Fig. 13). The amount of dual FAK-paxillin containing FAs was also initially increased by defactinib treatment followed by a decrease at higher concentrations; the trend resulted in an IC$_{50}$ value of 11.17 μM (Fig. 5j and Supplementary Fig. 13). Overall, these experiments showed that our lead FAK FAT inhibitor 2012 disrupts FAK localization with high nanomolar potency, whereas traditional ATP-competitive FAK kinase domain inhibitors increase FAK-containing FAs.

## 2012 shows selective and robust anti-cancer effects

Because disruption of the FAK FAT domain via FRNK has shown the ability to selectively reduce cancer cell viability[39], we next examined the effects of 2012 and defactinib treatment after 96 h on cellular viability and cancer cell selectivity. Treatment with 2012 did not produce a dose-dependent reduction in the viability of normal human epidermal melanocyte (NHEM) cells, normal human hepatic LX-2 cells, or human hepatoma Hep-G2 cells (Fig. 6a). However, in melanoma cell lines SK-MEL-147 and WM88, as well the glioblastoma cell line ONDA7, treatment with 2012 yielded low micromolar IC$_{50}$ values of 2.36, 4.42, and 9.79 μM, respectively; showing 2012 can selectively reduce cancer cell viability (Fig. 6a). To further validate the decrease in cell viability was due to selective disruption of the FAK-paxillin interaction and not non-specific membrane damage, LDH assays were performed, and no significant levels of cytotoxicity were observed (Supplementary Fig. 17). Treatment with defactinib was also able to reduce the viability of cancer cells SK-MEL-147, WM88, and ONDA7 but the effects were less potent compared to those of 2012 with higher IC$_{50}$ values of 2.84, 21.18, and 18.89 μM respectively (Fig. 6b). Additionally, defactinib also decreased the viability of normal human hepatic LX-2 cells with a low micromolar IC$_{50}$ of 2.19 μM (Fig. 6b). Treatment with inactive control peptide 2020 did not reduce the viability of any cell line tested (Fig. 6c).

Recognizing kinase inhibitors often have many off-target effects due to targeting the highly conserved ATP-binding pocket and the observed decrease in viability following treatment with defactinib in non-cancerous hepatic LX-2 cells, we next investigated the kinome selectivity of FAK kinase inhibitor, defactinib. The KINOMEscan performed by Eurofins (Supplementary Table 5) showed that defactinib (1 μM) inhibited 45 out of 97 kinases tested at a level >50%, most

prominently FAK, FLT3, JAK2/3, AURKA, TRKA, and TYK2; and resulted in selectivity scores for S(1), S(10), and S(35) of 0.033, 0.122, and 0.256, respectively (Fig. 6d). As expected for a PPI inhibitor, 2012 (1 μM) inhibited 0 out of 97 kinases tested >50%, resulting in a selectivity score of 0 for S(1), S(10), and S(35), emphasizing the selectivity advantage of targeting the FAT domain over the kinase domain.

Because traditional FAK kinase domain inhibitors have limited effects on cancer cell apoptosis[12,13] but disruption of the FAT domain induces apoptosis[17–19], we next tested lead FAT inhibitor 2012 in annexin V apoptosis assays. Peptide 2012 had an EC$_{50}$ of 8.3 μM at 96 h, whereas defactinib did not exhibit a dose-dependent effect (Fig. 6e). In time-course experiments, 2012 induced a dose- and time-dependent induction of apoptosis, whereas 2020 (negative control) and defactinib had no effect at any timepoint (Supplementary Fig. 18).

In vitro assays were then performed to evaluate the effect of 2012 on cancer cell invasion, a prominent oncogenic process mediated by the FAK FAT domain. Peptide 2012 showed a dose-dependent decrease in cancer cell invasion compared to vehicle control DMSO, with an IC$_{50}$ of 1.8 μM (Fig. 6f). Negative control peptide 2020 had no effect on SK-MEL-147 invasion. Defactinib had no effect on cell invasion until 2.5 μM, where the percentage of invaded cells sharply dropped to 0% (Supplementary Fig. 19), indicating potential toxicity beyond the 2.5 μM dosage and poor target specificity. In all, these experiments indicated that FAT-targeted peptides have anti-FAK biological effects, such as dose-dependent induction of apoptosis and inhibition of invasion, and offer increased selectivity compared to traditional FAK kinase domain inhibitors.

## 2012 displays favorable DMPK properties and in vivo efficacy

To ensure suitability for in vivo testing, a series of in vitro DMPK assays were performed to assess plasma and metabolic stability of 2012. Peptide 2012 had high plasma stability ($t_{1/2}$ > 120 min) vs. control molecule propantheline bromide ($t_{1/2}$ = 66 min; Supplementary Fig. 20a) and had increased metabolic stability in human liver microsomes ($t_{1/2}$ > 60 min) vs. positive control molecule imipramine ($t_{1/2}$ = 22 min; Supplementary Fig. 20b). Given this promising in vitro DMPK, in vivo pharmacokinetic analysis was performed in Balb/cJ mice following single intraperitoneal (IP) administration; 2012 exhibited a long plasma half-life of 16.5 h (Fig. 6g), low plasma clearance (0.26 mL/min/kg), and C$_{max}$ of 89.4 μg/mL (48.5 μM molar concentration), which was 60-fold excess of 2012 plasma concentration compared to required in vitro therapeutic levels (focal adhesion IC$_{50}$ = 0.8 μM).

Next, the ability of 2012 to block tumor growth was evaluated in the B16F10 syngeneic mouse model of melanoma, in which tumor growth was shown to be sensitive to FAT domain mutation[40]. Intraperitoneal treatment with peptide 2012 at a daily dose of 20 mg/kg resulted in a statistically significant reduction in tumor burden (80.2%) compared to vehicle-only control (Fig. 6h). Furthermore, peptide 2012 was well tolerated for the duration of the study, evidenced by the lack of body weight fluctuation and no observable signs of toxicity (Supplementary Fig. 20c). These data demonstrated that pharmacological FAT inhibition has a pronounced anti-tumor effect with proper safety. This recapitulates the observation that 2012 reduced only cancer cell viability and did not impact normal cells in vitro.

## Discussion

FAK is an attractive target for cancer therapeutics, but existing FAK-targeting drugs focus solely on inhibition of the kinase domain. In contrast, the FAT domain scaffold forms critical interactions with focal adhesion proteins and oncogenic receptor tyrosine kinases, mediating multiple signaling cascades for tumorigenesis, providing a potentially safe and effective therapeutic advantage over traditional kinase inhibition (Supplementary Fig. 21).

This report describes the structure-based discovery of inhibitors of the FAK FAT domain utilizing the native sequence of the alpha-

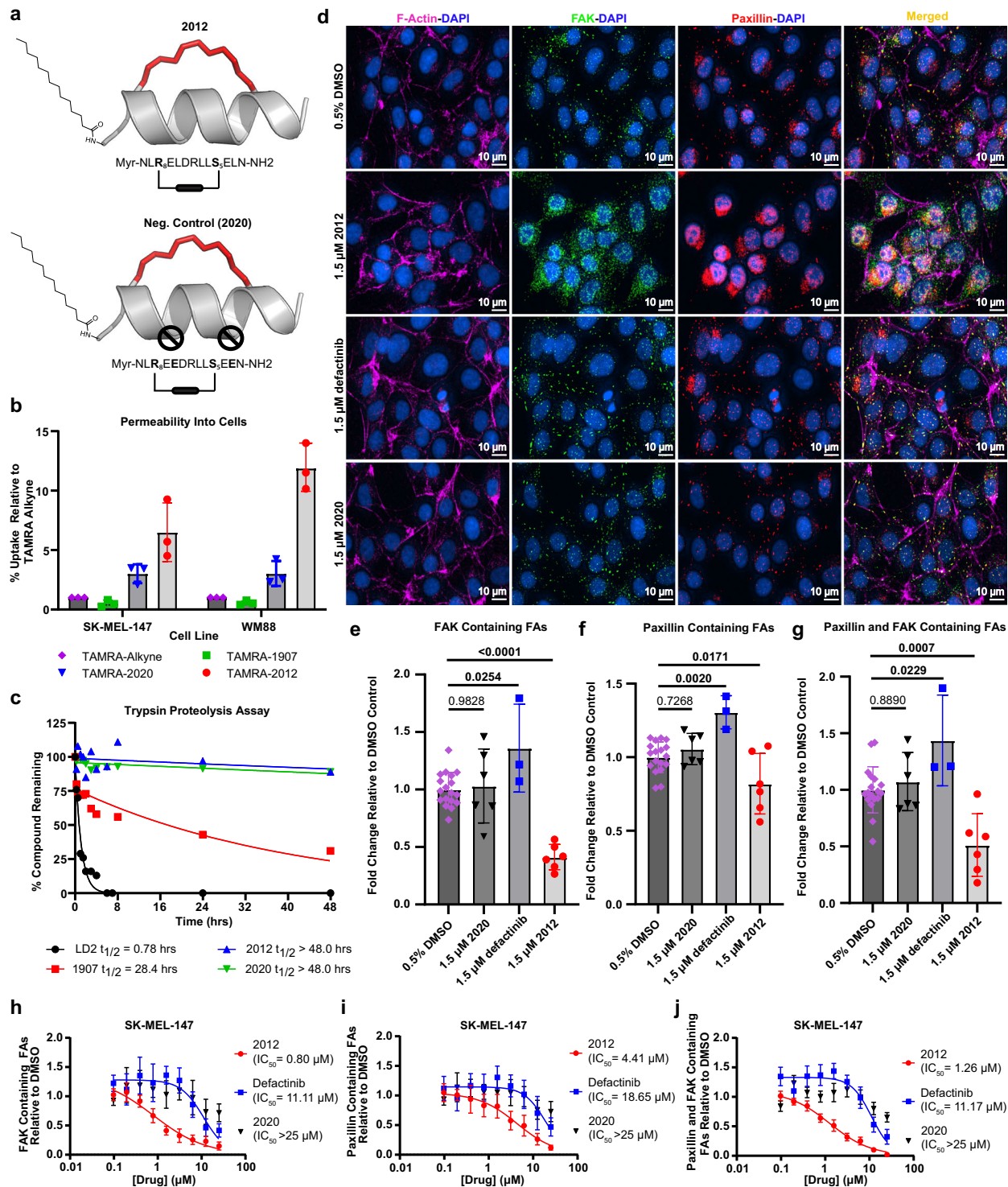

helical LD2 motif of paxillin (conserved sequence: SELDXLLXXL). Analysis of the X-ray co-crystal structure of FAT-LD2 complex[24] and residue by residue SAR facilitated the rational design of a peptide (i.e., 1907) with >100-fold improved binding compared to the native LD2 sequence. We synthesized multiple peptides that varied in staple length and position. Through this empirical SAR process, we identified 1907, with an $i, i + 7$ staple in positions S143 and L150 facing directly away from the binding interface, that outperformed other analogs in binding and competition assays. All other efforts ($i + 3, i + 4$, Aib, etc.), including analogs with staples oriented at the hydrophobic interface, did not provide peptides with comparable activity. These findings

highlight the customization required for the activity of 1907, where a single staple length and position was necessary for optimal activity. The fact that a solvent-exposed hydrocarbon stapling motif (1907) provided the greatest increase in FAK FAT binding was unexpected, as another stapled peptide drug candidate ATSP-7041, which targets the p53-MDM2/MDMX interaction[34], demonstrates high-affinity binding through a staple on the lateral side of the alpha-helix. Further, SAR experiments demonstrated the sequence specificity required for 1907 activity. Particularly, negatively charged residues E144 and E151 and residues located N-terminal (N141, L142) and C-terminal (E151, L152, and N153) to the staple on 1907 were critical for activity. We also

**Fig. 5 | Peptide 2012 exhibits successful cell permeability and disruption of focal adhesions, demonstrating differential anti-FAK biological effects in vitro.**
**a** Peptide 2012 (top) is an analog of the stapled peptide 1907, with a myristoyl group attached to the N-terminus. Peptide 2020 (bottom) serves as an inactive negative control for 2012, as it contains two inactivating residue substitutions hindering its interaction with the FAK FAT binding sites. **b** Fluorescence-based permeability assays in SK-MEL-147 and WM88 human melanoma cells showing cellular uptake of 10 μM TAMRA-tagged 2012, unmyristoylated TAMRA-tagged 1907, and TAMRA-tagged 2020 relative to the permeability of TAMRA-alkyne control after 48 h. Bar graphs depict the mean ± SD of three biological replicates (n = 3). **c** Trypsin proteolytic resistance assay displaying half-lives for LD2, 1907, and 2012. **d** Immunofluorescence staining of F-actin (magenta), FAK (green), paxillin (red), and DAPI (blue) in SK-MEL-147 melanoma cells after 48 h of treatment with 1.5 μM

2012, defactinib, and 2020 (neg. control) along with 0.5% DMSO (vehicle) control and corresponding changes in **e** FAK, **f** paxillin, and **g** dual FAK-paxillin containing focal adhesions. Bar graphs report mean change in FA composition ± SD. One-way ANOVA analyses and Dunnett's multiple comparison tests were performed with corrected p values provided. Bolded p values indicate statistically significant values where $p < 0.05$. Sample size was 18 (n = 18) for DMSO (vehicle control); 6 (n = 6) for 2012 and 2020 treated groups; and 3 (n = 3) for defactinib treated groups where "n" represents independent biological replicates. Dose-response curves showing the mean ± SD reduction of **h** FAK, **i** paxillin, or **j** dual FAK-paxillin containing focal adhesions and corresponding $IC_{50}$ values after 48 h of treatment with 2012, defactinib, or 2020 relative to 0.5% DMSO control in SK-MEL-147 cells. All groups had three biological replicates (n = 3). All source data are provided as a Source Data file.

determined that the hydrophobic leucine residues in 1907 (L142, L145, L148, L149, and L152) are buried deeper into the FAT domain's leucine binding pocket, up to 1.64 Å closer than seen in the LD2 model, which was an unexpected finding and may explain the increased affinity of 1907. Substitution of these critical leucine residues to tryptophan eliminated binding affinity, highlighting the importance of maintaining key hydrophobic contacts of leucine side chains, and not general hydrophobicity, which is further exemplified by the evolutionary conservation of leucine residues in LD motifs[41]. Overall, structural modifications, including the stapling motif (i,i + 7), specific positioning of the staple, N- and C-terminal residues, conservation of leucine side chains, and preservation of charged/polar residues D146, E151, and N153 directed the improved potency and specificity of 1907 as a FAK scaffold inhibitor.

Myristoylation of 1907 (analog 2012) drastically increased cellular permeability and anti-cancer effects. In cellular models, 2012 was able to potently and effectively reduce the number of FAK-containing focal adhesions with a sub-micromolar $IC_{50}$ value of 0.80 μM, which was on par with the $K_D$ of 0.8 μM at the helix 2–3 FAT domain site. Given this is a >100-fold improvement compared to the native LD2 sequence ($K_D = 82.6$ μM), these data suggest that 2012 has sufficient potency to effectively disrupt FAK localization to focal adhesions, whereas low nanomolar potency typical of ATP-competitive kinase inhibitors is not necessary. Furthermore, our data suggest potent inhibition of the FAT helix 2-3 site is sufficient for cellular disruption of the FAK-paxillin interaction, which is in alignment with previous reports characterizing the interaction[25]. The amount of paxillin containing focal adhesions was reduced only at higher concentrations of 2012, with an $IC_{50}$ value sixfold larger. This discrepancy may be due to paxillin's inherent FA targeting through the four highly conserved LIM domains at its C-terminus, which localize the protein to FAs by either direct interaction with β-integrin tails or other intermediate binding partners[42]. However, the order of protein localization to FAs is controversial, with conflicting reports showing in some cases that paxillin localization is recruited by FAK, which may account for 2012's observed reduction in paxillin and dual FAK-paxillin containing FAs[43]. Nonetheless, our data suggest that paxillin primarily recruits FAK to focal adhesions, which is dose-dependently inhibited by peptide 2012.

Phosphorylation of paxillin at Y118 by FAK has been shown to promote the disassembly and turnover of focal FAs[44]. Blocking this process with FAK kinase inhibitors may stabilize sites of focal adhesions and explain the increase in FAK, paxillin, and dual FAK-paxillin containing focal adhesions we observed following treatment with high nanomolar to low micromolar doses of defactinib. The disruption of focal adhesion complexes by 2012 led to robust phenotypic changes in cancer cells, a key difference from FAK kinase inhibition; defactinib did not induce cancer cell apoptosis and inhibited cell invasion with a very steep Hill coefficient, presumably due to non-specificity or toxicity[45]. Peptide 2012 also showed significant selectivity for cancer cells over normal cells, whereas defactinib was non-selective, as evidenced by the inhibition of 44 additional kinases other than FAK after 1 μM of

treatment. This is in alignment with other reports showing the non-selectivity of FAK kinase inhibitors and inhibition of cell growth in an FAK-independent manner[13,46]. Conversely, previously published studies with genetic FAK FAT domain inhibitor, Ad-FRNK, reported selective inhibition of cancer cell viability and limited effects on normal cell viability[19,39]. Furthermore, the FAT domain of FAK has limited sequence similarity to other proteins contained in the proteome, and therefore direct FAT ligands are predicted to have a potential selectivity advantage over traditional ATP-competitive kinase domain inhibitors, which is supported by the kinome profiling of 2012 (Fig. 6d). Future work will investigate the differential mechanism of action of 2012 compared to traditional kinase domain inhibition, particularly investigating the mechanism by which kinase inhibitors are able to reduce cancer cell viability. Previous precursors have shown off-target inhibition of cyclin-dependent kinases (CDKs)[12,13], and other reports have indicated FAK kinase inhibitors may induce cancer cell senescence[47] rather than apoptosis; however, further validation is required.

Peptide 2012 demonstrated promising pharmacological effects, as observed in in vitro DMPK, in vivo pharmacokinetics, and in vivo efficacy studies. Remarkably, 2012 showed no proteolytic cleavage even after 48 h incubation, confirming the anticipated benefit of the hydrocarbon stapling moiety on chemical stability and resistance to proteases[32]. The pharmacokinetic data of peptide 2012 was also compelling, with a half-life ($t_{1/2}$) of 16.5 h in mice after a single IP administration. Because drug pharmacokinetics ($t_{1/2}$) in mice are often much faster than in primates or humans[48], this value represents an excellent benchmark for future drug development efforts. As a reference, stapled peptide drug candidate ATSP-7041, which targets the p53-MDM2/MDMX interaction and is a precursor of clinical lead ALRN-6924, has a plasma half-life of 1.5 h in mice and 18.3 h in monkeys[34]. Furthermore, in the highly aggressive B16F10 mouse tumor model, which progresses from palpable tumors to >2000 mm³ in 2 weeks, the tumor burden was significantly reduced by 2012. To calculate tumor volume in this initial study, we utilized the simplified ellipsoid volume formula (Eq. 4), which is commonly used in the field to estimate tumor volume but does not directly measure tumor depth; however, all tumor volumes were calculated using the same formula and thus support the conclusion that 2012 reduced tumor volume. Initial in vitro and in vivo pharmacology studies were focused on melanoma due to its established link with FAK signaling. FAK is activated in late-stage cutaneous and uveal melanoma[49], and inhibition of the FAT domain using Ad-FRNK[50] or mutagenesis (Y925F)[40] inhibits melanoma growth, invasion, and metastasis. While our initial dataset focused only on primary tumor burden, future melanoma studies will include testing in additional models, i.e., patient-derived xenograft models with clinically relevant genetic alterations (e.g., *BRAF*, *MEK*, and *NRAS* mutations), metastasis models, and syngeneic models in conjunction with immunotherapy (e.g., anti-PD-1, anti-CTLA-4). Additionally, we will further explore the potential of FAK FAT PPI inhibitors in other indications, including

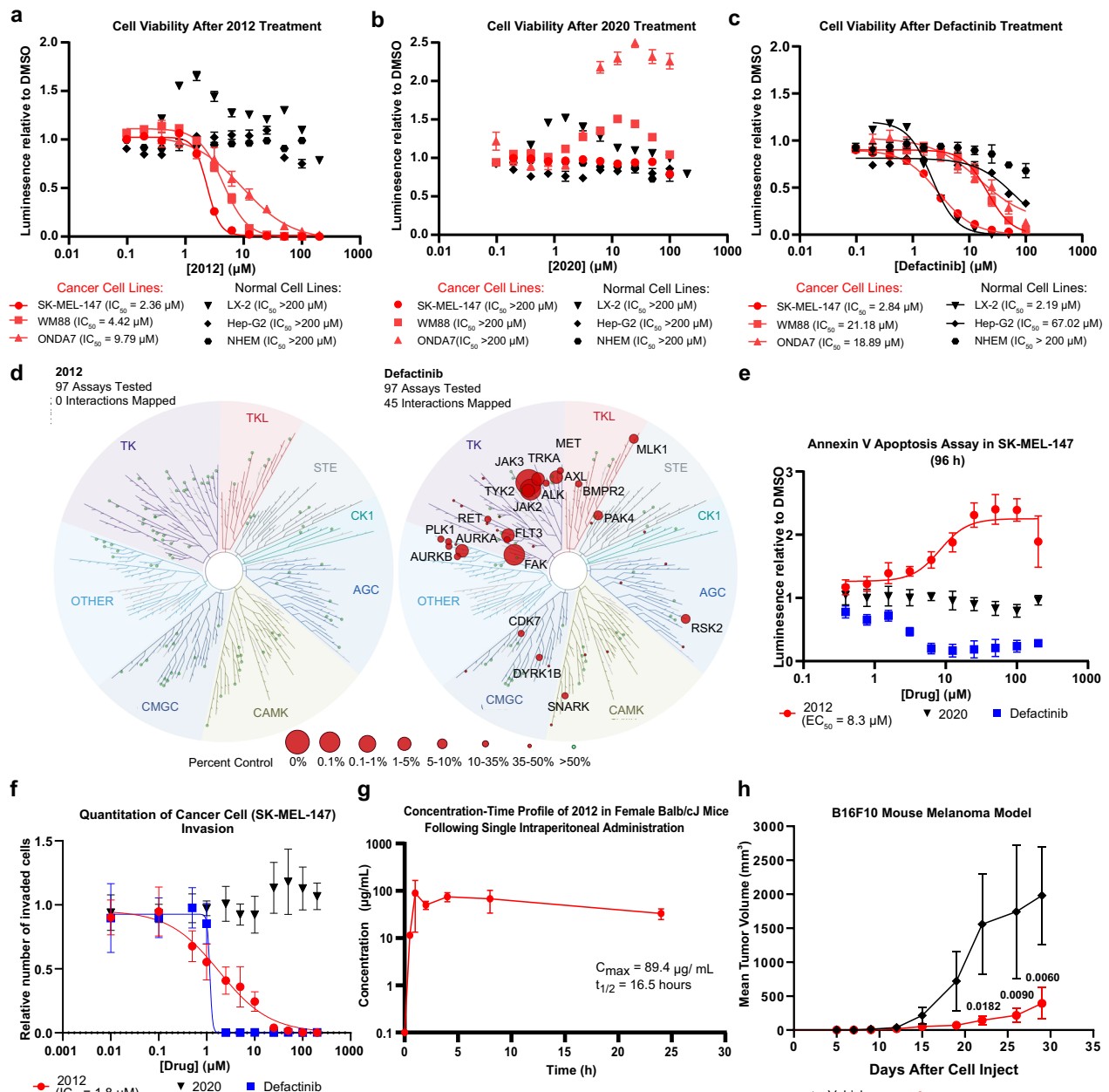

**Fig. 6 | Peptide 2012 induces robust anti-cancer effects in both in vitro and in vivo cancer models.** Dose-response curves reporting mean ± SD decrease in cell viability and corresponding average $IC_{50}$ values of cancer cell lines SK-MEL-147, WM88, and ONDA7, as well as normal cell lines LX-2, Hep-G2, and NHEM, treated with serially diluted **a** 2012, **b** defactinib, and **c** 2020 (neg. control) for 96 h. Values reported for: SK-MEL-147 and LX-2 were derived from 4 biological replicates ($n = 4$) in (**a**–**c**); WM88 were derived from 4 biological replicates ($n = 4$) in (**a**) and 3 biological replicates in (**b**, **c**); ONDA7 were derived from 5 biological replicates ($n = 5$) in (**a**, **c**) and 3 biological replicates ($n=3$) in (**b**); Hep-G2 were derived from 3 biological replicates ($n = 3$) in (**a**–**c**); and NHEM were derived from 3 biological replicates ($n = 3$) in (**a**) and 4 biological replicates ($n = 4$) in (**b**, **c**). **d** Selectivity profile of 2012 and defactinib at 1 μM against 97 kinases measured in the KINO-MEscan assay (Eurofins) where the size of red circles represents the percent of remaining kinase activity relative to DMSO control. **e** Annexin V Apoptosis $EC_{50}$ curves reporting mean luminescence relative to DMSO (vehicle control) ± SD following treatment with 2012 ($n = 7$); 2020 ($n = 5$); or defactinib ($n = 5$) obtained after

96 h of drug treatment. All $n$'s represent independent biological replicates. **f** Invasion $IC_{50}$ curves reporting mean ± SD reduction of invasion relative to DMSO (vehicle control) obtained from transwell invasion chamber assays after 48 h treatment of 2012 ($n = 6$), 2020 (neg. control) ($n = 2$), and defactinib ($n = 4$) in SK-MEL-147. All $n$'s represent independent biological replicates. **g** Pharmacokinetic evaluation via i.p. administration of 2012 in Balb/cJ mice revealing $C_{max}$ and plasma half-life. The concentration-time profile reports the remaining mean concentration of 2012 (μg/mL) ± SD at 0, 0.5,1, 2, 4, 8, and 24 h derived from 3 biological replicates ($n = 3$). **h** Effects on tumor growth in B16F10 syngeneic melanoma mouse model by 2012 treatment (20 mg/kg i.p. QD, $n = 8$ mice per group). Treatment started at day 12. Curves report mean tumor volume ± SD derived from 8 biological replicates per group ($n = 8$). Two-way ANOVA analysis and Dunnett's multiple comparison test were performed, and statistically significant changes in tumor volume were denoted with corrected $p$ values. Bolded $p$ values indicate statistically significant values where $p < 0.05$. All source data are provided as a Source Data file.

idiopathic pulmonary fibrosis, which has been shown to be negatively regulated by Ad-FRNK[9].

Along with the differential strategy of targeting the FAT domain, hydrocarbon stapling optimizes the peptide drug discovery approach

and provides an advantage over small molecule drugs. As demonstrated with peptide 2012, the hydrocarbon staple improves helicity and maintains the stability of protein mimetics to confer improved functionality. Both the staple and the myristoyl group are also known

to aid in cellular permeability, allowing enhanced efficacy in vitro, in vivo, and potentially in the clinic. Advancement of these stapled peptides has important implications for the success of this approach in targeting PPIs for treating cancer and other complex diseases such as pathogenic fibrosis, offering many advantages over small molecule inhibitors.

In conclusion, this report presents hydrocarbon-stapled paxillin-mimetic peptides for FAT inhibition and the X-ray co-crystal structure of a FAK scaffold inhibitor bound to its cognate target domain. Over the past 20 years, FAK has remained an attractive but elusive drug target for cancer therapy. With further development efforts, such peptide may yield the promise of a monotherapy successfully targeting FAK in cancer and other relevant clinical conditions. Together, these results represent a major leap forward in the FAK field in the development of a stapled peptide inhibitor to target the FAT domain.

## Methods

### Research compliance
The mouse studies are compliant with all relevant ethical regulations for animal experiments. All procedures were approved by the University of Arizona Institutional Animal Care and Use Committee (IACUC). Euthanasia protocols upon tumor volumes >2000 mm$^3$ were followed in accordance with the University of Arizona IACUC.

### Synthesis and HPLC of hydrocarbon-stapled peptides
Solid phase peptide synthesis (SPPS) was performed using N-α-fluorenylmethoxycarbonyl (Fmoc) protected amino acids. All peptides were synthesized on a 0.1–0.4 mmol scale with a Biotage Initiator+ Alstra automated microwave-assisted peptide synthesizer using H-Rink Amide ChemMatrix resin (0.45 meq/g loading). Hydrocarbon-stapled peptides were synthesized via the inclusion of two olefinic amino acids, followed by first-generation ruthenium-catalyzed olefin metathesis, similarly as described[30]. Standard coupling, deprotection, and cleavage conditions for SPPS are detailed in Supplementary Fig. 1.

Analytical HPLC was performed on an Agilent 1200 HPLC, with 1100 DAD and Infinity 6125 LC-MS detectors. Separations were achieved on Zorbax SB-C18 (Agilent 830990-902; 2.1 × 150 mm; 80 Å pore size; 3.5 μm particle size), and Luna C8(2) (Phenomenex 00F-4249-B0; 2 × 150 mm; 100 Å pore size; 5 μm particle size) columns with gradients of either acetonitrile (0.1% AcOH) in water (0.1% AcOH) or methanol (0.1% NH$_4$OAc) in water (0.1% NH$_4$OAc). All tested compounds had purities ≥ 95% at 214 nm. Mass spectra were consistent with the structures (see Supplementary Table 1). OpenLab CDS ChemStatio (version 3.0) software was used for LC-MS and prep-HPLC.

Preparative HPLC was performed on an Agilent 1260 II quaternary HPLC / Variable Wavelength Detector and automatic collector. Separations were achieved on Zorbax SB-C18 (Agilent 880975-202; 9.4 × 250 mm; 80 Å pore size; 5 μm particle size) and Luna C8(2) (Phenomenex 00G-4249-N0; 10 × 250 mm; 100 Å pore size; 5 μm particle size) columns with gradients of either acetonitrile (0.1% AcOH) in water (0.1% AcOH) or methanol (0.1% NH$_4$OAc) in water (0.1% NH$_4$OAc).

### Surface plasmon resonance (SPR)
SPR binding studies were performed using biotinylated FAK FAT protein[51]. Briefly, SPR binding assays were performed on a ForteBio Pioneer FE SPR system using SPR running buffer (100 mM Tris-Cl pH 8.0, 200 mM NaCl, and 0.05% Tween-20). Biotinylated AviTag FAT protein was immobilized on a Streptavidin in Dextran Hydrogel (SADH) biosensor (ForteBio). Desired immobilization levels were determined utilizing the following Eq. 1:

$$R_{max} = R_{protein} \, x \left( \frac{Mr_{analyte}}{Mr_{protein}} \right) x V_{protein} \qquad (1)$$

where $R_{max}$ is the highest possible response of binding between the analyte and the protein, $R_{protein}$ is the change in response by the immobilization of the protein, $M_r$ represents the mass of the analyte or protein, and $V_{protein}$ represents the number of binding sites on the protein. A target $R_{max}$ of approximately 100 RU was chosen.

All samples were diluted in the SPR buffer to a volume of 200 μl and injected using the OneStep gradient injection system at a flow rate of 150 μl/min. For a bulk standard control, a 3% (w\v) solution of sucrose in SPR buffer was used for injection. A DMSO calibration curve was created as well using a concentration range from 3.5 to 6.5% DMSO. Peptides were titrated from 200 μM to 97 nM. SPR data analysis utilized Qdat software (ForteBio, version 4.3.2 build 8) for normalization of the baseline prior to injection, channel alignment, and reference channel and blank subtraction. All peptides fit to a pseudo first-order one-site binding model to calculate the $K_D$ from kinetic data, other than peptides 1907 and 1920, which fit to a two-site binding model.

### Fluorescence polarization (FP)
Fluorescence polarization competition (FAK-paxillin) assay was performed using FAK FAT protein and TAMRA-LD2 L10D peptide[52]. Briefly, FP reaction consisted of FP buffer (20 mM Tris-Cl pH 8.0, 200 mM NaCl, 0.05% β-mercaptoethanol, 0.1% Triton-X 100, 5% glycerol and 1X HALT protease inhibitor cocktail (Thermo)), FAT protein (25 μM), TAMRA-LD2 L10D probe (0.1 μM), and peptide/DMSO (6.25%). Peptides were titrated from 1.25 mM to 610 nM, and the reaction was agitated on a shaker for 3 h. The plate was read on a BMG LabTech ClarioStar Plus plate reader using BMG ClarioStar MARS software (version 4.20) with a 540 nm excitation filter, a 590 nm emission filter, and a 565 nm dichroic filter. GraphPad Prism (version 10.3.1) was used to produce a dose-response curve and determine the IC$_{50}$. The IC$_{50}$ was calculated using a four-parameter dose-response inhibition model, with the bottom fit constrained to the lower plateau of the binding curve. The K$_i$ was calculated from the following Eq. 2:

$$K_i = \frac{I_{50}}{\frac{L_{50}}{K_D} + \frac{P_0}{K_D} + 1} \qquad (2)$$

where $I_{50}$ is the concentration of free peptide at 50% inhibition, $L_{50}$ is the concentration of FAT at 50% inhibition, $P_O$ is the concentration of free protein, and $K_D$ is calculated from the saturation curve of FAT and TAMRA-LD2 L10D.

### Purification of $^{15}$N labeled FAK FAT
[U-$^{15}$N]-labeled FAT (residue S892-H1052) was produced in *E. coli*[51]. Briefly, FAT protein was expressed using minimal media with M9 salts: 6.8 g/L Na$_2$HPO$_4$, 3.0 g/L KH$_2$PO$_4$, 0.5 g/L NaCl, 1 g/L $^{15}$NH$_4$Cl, 2 mM MgSO$_4$, 100 μM CaCl$_2$, 0.4% glucose, 500 μM thiamin, 0.5X MEM vitamins, and 10 μM FeCl$_3$ • 6H$_2$O, CuSO$_4$ • 5H$_2$O, MnSO$_4$ • H$_2$O, and ZnSO$_4$ • 7H$_2$O. Purified $^{15}$N FAT was buffer exchanged into thrombin cleavage buffer (50 mM Tris-Cl pH 8.0, 10 μM CaCl$_2$) on a HiLoad 26/60 Superdex 75 pg gel filtration column. Buffer-exchanged FAT was incubated with thrombin beads (Sigma-Aldrich) at a concentration of 1 mg/mL for 6 h at room temperature. Beads were removed by centrifugation (500×*g* for 2 min), and the supernatant was collected and incubated with Ni-NTA resin to remove un-cleaved protein and remaining His tag. The cleaved protein was gel filtered through a HiLoad Superdex 75 pg 26/60 gel filtration column into NMR buffer (100 mM Tris pH 8.0, 100 mM NaCl, 2 mM DTT, 50 μM DSS, 0.02% sodium azide, and 5% D$_2$O).

### Protein NMR
NMR spectra were collected on a Bruker Avance III-HD console operating at Larmor frequencies of 600.133 and 60.817 MHz for $^1$H and $^{15}$N, respectively. The spectrometer was equipped with a 5 mm TCI Prodigy

cryoprobe. $^1$H-$^{15}$N TROSY-HSQC (*trosyetf3gpsi*) spectra were collected with spectral widths of 14.03 and 30.0 ppm, 2048 and 256 points, respectively, and 16 scans per increment. NMR data were collected with Topspin 3.6.5 and 25C. NMR resonance assignments of FAT were taken from prior experiments[51] (BMRB 28012) and matched with identical buffer conditions. NMR studies were performed in NMR buffer 100 mM Tris pH 8.0, 100 mM NaCl, 2 mM DTT, 50 µM DSS, 0.02% sodium azide, and 5% D$_2$O) and 1% DMSO across all concentration points. Previous backbone assignments were taken at 1% DMSO and the same NMR buffer. Peptide 1907 was screened at 200, 100, 50, 25, 10, 5, 1, 0.5, 0.1, and 0 µM with 100 µM $^{15}$N-labeled FAT. All spectra were processed and analyzed using Mestrelab Research MestReNova (version 14.2.0) and Bruker TopSpin (version 3.6.3). For Kd estimation, a slow-exchange binding model was utilized, allowing for the evaluation of residue-specific isotherm curves by tracking changes in peak intensity during the titration. We computed binding curves by least squares regression, using the Eq. 3[35]:

$$\Delta V = \frac{[P] + [L] + K_d - \sqrt{([P] + [L] + K_d)^2 - 4[P][L]}}{2[P]} \quad (3)$$

Where $\Delta V$ is one minus the ratio of peak volume relative to its volume in the unbound protein spectrum, expressed as a decimal, *[P]* is the concentration of protein, and *[L]* is the concentration of ligand in the sample. Binding isotherms were plotted using GraphPad Prism (version 10.3.1) using the one site-specific binding least squares fit with standard error.

### Purification of recombinant FAT protein for crystallography
The human FAK FAT domain (residues R919-H1052, referred to as FAT919) was expressed and purified from BL21(DE3) *E. coli*[53]. Bacteria were grown in TB broth at 37 °C to an OD$_{600}$ of 1.2, and protein expression was induced with the addition of 0.2 mM of isopropyl β-ᴅ-1-thiogalactopyranoside (IPTG). Bacteria were further incubated at 37 °C for 4 h before being centrifuged at 5000×*g* for 10 min. Cell pellets were collected and resuspended in lysis buffer (50 mM Tris pH 8.0, 200 mM NaCl, 5 mM β-mercaptoethanol, 10% glycerol, and 1X Halt protease inhibitor (Thermo Fisher), and frozen overnight at −80 °C.

After thawing, cell pellets were sonicated and centrifuged for 15 min at 12,000×*g* to remove insoluble fractions. The supernatant was collected and incubated with Glutathione Sepharose resin (Thermo Fisher) overnight to allow binding of the GST-tagged FAT protein. The mixture was spun down at 1500×*g* for 5 min to separate the resin-protein complex. The resin was washed with lysis buffer without 1X HALT, then eluted with reduced glutathione elution buffer (lysis buffer with added 50 mM reduced glutathione). Elutions were pooled, and the GST tag was removed from FAT via Thrombin Cleavage Kit (Sigma) into 50 mM Tris pH 8.0, 200 mM CaCl$_2$. The sample was further purified via size exclusion using a HiLoad 26/60 Superdex 75 pg gel filtration column and equilibration buffer (20 mM Tris pH 8.0, 150 mM NaCl, 5 mM β-mercaptoethanol). Data collection and analysis performed in Unicorn (version 5.0).

### Crystallography and structure determination
FAT919 (1.3 mM) and 1907 (5 mM) were mixed in a 1:1.5 (v/v) ratio in 20 mM Tris pH 8.0, 150 mM NaCl and incubated overnight at 4 °C. The protein:ligand complex was crystallized after 3 days of incubation at 20 °C in 20 mM zinc chloride and 20% (w/v) PEG 3,350 by the hanging-drop vapor diffusion method (2 µL protein-ligand complex:2 µL crystallization solution). Crystals were transferred and soaked in 2 µL droplets of crystallization solution for 5 min to ensure sufficient incorporation of cryo-protectant (20% (w/v) PEG 3,350), and subsequently flash frozen in liquid nitrogen for storage. Data were measured on beamline 23-ID-D at the Advanced Photon Source using standard collection parameters: wavelength, 1.033 Å; temperature, 100 K. Data

were processed using XDS[54] (2019 version) for indexing and the CCP4 Program Suite[55] (version 7.0) for scaling and merging. The structure was solved by molecular replacement with coordinates and phases derived from the FAT-LD2 complex (PDB ID 1OW8[24]) as an initial search model and then refined in PHENIX[56]. Data collection parameters and refinement statistics are reported in Supplementary Table 2. The final structure had a Rwork/Rfree (%/%) of 21.72/24.95 with no Ramachandran or Rotamer outliers. Figures and distance measurements were made in PyMOL[57] (version 1.8). The final structure was deposited in the Protein Data Bank (6PW8).

### Circular dichroism (CD) spectroscopy
CD experiments were performed on a Jasco J-815 CD spectrometer equipped with a Jasco PTC-423S/15 temperature control and analysis utilizing Spectra Manager (version 2.0). Spectral measurements were made in quartz spectrophotometer cells with an optical cell path length of 1.0 mm (Starna Cells Inc.) using standard collection parameters: 20 °C, 190–260 nm, 0.5 nm step resolution, 50 nm/min scanning speed, 0.5 s response, 1 nm bandwidth, 8 accumulations. Lyophilized peptides were reconstituted in 20 mM NaH$_2$PO$_4$ pH 7.0, and concentrations (63.3 µM for LD2, 60 µM for 2013, and 59.7 µM for 1907) adjusted to control the high-tension voltage (HT[V]) to ensure reliable data up to 190 nm. Secondary structure analysis, including % alpha-helicity, was performed using the CDSSTR method[58] on data reference set 4 basis spectra. The reliability of secondary structure predictions was quantified by the goodness-of-fit of the back-calculated spectrum to the experimental spectrum and reported as a normalized root-mean-square deviation (NRMSD).

### Cell culture
The human melanoma cell lines SK-MEL-147 (Memorial Sloan Kettering Cancer Center) and WM88 (Rockland, Cat #WM88-01-0001), the glioblastoma cell line ONDA7 (JCBR, Cat #JCBR1575), the breast cancer cell line SK-BR-3 (ATCC, Cat #HTB-30), the human kidney cell line HEK293T (ATCC, Cat #CRL-3216), and the mouse melanoma cell line B16F10 (ATCC, Cat #CRL-6475) were cultured in high glucose DMEM (Gibco), fortified with 10% syringe-filtered FBS (Gibco), 1% penicillin/streptomycin (Gibco), and 0.2% Normocin (InvivoGen). Normal hepatic stellate cells, LX2 (Millipore Sigma, Cat #SCC064), were cultured in DMEM plus 2% FBS. Human hepatoma Hep-G2 cells (ATCC, Cat #HB-8065) were cultured in EMEM containing 2 mM ʟ-glutamine, 1 mM sodium pyruvate, and 1500 mg/L sodium bicarbonate (Gibco), fortified with 10% syringe-filtered FBS (Gibco), 1% penicillin/streptomycin (Gibco), and 0.2% Normocin (InvivoGen). Normal human epidermal melanocytes (PomoCell, Cat #C-12413) were cultured in Melanocyte Basal Medium (PomoCell), including Melanocyte Growth Medium SupplementMix (PomoCell). All cells were grown at 37 °C in 5% CO$_2$. All cells were authenticated by the University of Arizona Genetics Core.

### Cell viability assay
Cells were seeded at 1000 cells per well in white-walled, clear bottom 96-well plates and allowed to adhere in the incubator for 24 h. Cells were then gently washed with Dulbecco's phosphate-buffered saline (DPBS) to remove serum-containing media and treated with peptide 2012, peptide 2020, or defactinib in triplicate replicates. Treatment doses were created in a 12-or 10-point, twofold serial dilution, starting at 200 µM. After 24 h of treatment, 10% FBS was spiked back into each well and the plate returned to 37 °C for 48 h. After 96 h of drug treatment, the Promega 2D CellTiter-Glo reagent was added to each well at a 1:2 ratio of reagent to treatment media. The plates were read on a BMG LabTech ClarioStar Plus plate reader using BMG ClarioStar MARS software (version 4.20) to detect luminescence as a readout for cell viability (% relative to DMSO). This assay was performed a minimum of 3 ($n$ = 3) times for each cell line, with each run containing a minimum of 3 technical replicates per run. IC$_{50}$ curves were generated using

GraphPad Prism (version 10.3.1), reporting the mean % viability relative to DMSO ± SD.

## LDH assay

Non-specific membrane disruption was assessed quantitatively by measuring the lactate dehydrogenase (LDH) activity in the medium using the cytoscan-LDH cytotoxicity assay kit (G-Biosciences Inc., St Louis, Mo, USA). SK-MEL-147 and WM88 human melanoma cells were plated in 96-well white-walled plates at 5k cells/well in 90 uL of serum complete DMEM and returned to a 37 °C incubator with 5% $CO_2$ overnight. The cells were then treated in triplicate with 2012, 2020 (negative control), and Gambogic Acid (positive control) at titrating concentrations ranging from 100 µM to 200 nM and allowed to incubate for 45 min in a 37 °C incubator with 5% $CO_2$. About 50 µl of supernatant was transferred from each well to a new flat-bottom 96-well white-walled plate and mixed with 50 µl of recombinant substrate mixture. This reaction was allowed to occur at room temperature for 30 min protected from light. Then, 50 µl of stop solution was added to terminate the reaction. Finally, plates were read on a BMG LabTech ClarioStar Plus plate reader using BMG ClarioStar MARS software (version 4.20) to detect absorption wavelengths at 490 and 680 nm. Results were reported as 490 nm values subtracted from the 680 values, averaged, three biological replicates were repeated per cell line, and graphs were generated using GraphPad Prism (version 10.3.1), reporting mean LDH activity ± SD.

## Immunofluorescence assay

SK-MEL-147 cells were collected and seeded into the inner 60 wells of 96-well, black-walled plates (Agilent, 204626-100) at 1000 cells per well in 100 uL of complete DMEM overnight in a 37 °C incubator with 5% $CO_2$. The next day, the serum-containing media was removed from each well and cells were washed with PBS before treating the cells in serum-free DMEM and 0.5% DMSO for 48 h. After treatment, cells were pre-fixed with 4% paraformaldehyde diluted in 1x PBS being added directly to the culture medium for 2 min. Pre-fixation media was then removed, and 4% paraformaldehyde diluted in 1X PBS was added to each well for 3 min at room temperature. Fixative was then removed and neutralized twice with 100 mM Tris Buffered Saline in MilliQ water and then washed once with 0.1% Tween-20 in PBS. Cells were permeabilized with 0.5% Triton-X 100 in PBS for 5 min at room temperature, then the permeabilization buffer was removed, and cells were washed twice with 0.1% Tween-20 in PBS. Cells were then blocked with 10% Goat Serum in PBS for 1 h at room temperature. Following blocking, cells were stained with pre-conjugated FAK 4.47 Ab- Alexa Fluor 555 (Millipore Sigma, 16-234) diluted 1:500 in Antibody Diluent Reagent Solution (Invitrogen, 003218) and Paxillin B2 Ab-Alex Fluor 647 (Santa Cruz Biotechnology, sc-365379 AF647) diluted 1:200 in Antibody Diluent Reagent Solution (Invitrogen, 003218) in the dark overnight at 4 °C. Antibody stain was then removed, and cells were washed twice with 0.1% Tween-20 in PBS. Actin staining was then performed with pre-conjugated Phalloidin-Alexa Fluor 488 (Cytoskeleton, PHDG1A) diluted 1:140 in Antibody Diluent Reagent Solution (Invitrogen, 003218) for 30 min in the dark at room temperature. Cells were then washed twice with 0.1% Tween-20 in PBS and DAPI stain (Sigma-Aldrich, D9542-5MG, 1 mg/mL) diluted 1:40,000 in PBS in the dark for 5 min at room temperature. DAPI stain was then removed, and cells were washed twice with 0.1% Tween-20 in PBS, then twice with PBS for 5 min per wash on an orbital shaker. Plates were then imaged on the BioTek Cytation C10 Confocal Imaging Reader. Three fields of view were taken per well per run and analyzed in the accompanying Agilent BioTek Gen5 Software (version 3.16). The following software parameters were used to quantify FAK-containing, paxillin-containing, and FAK/paxillin-containing focal adhesions: area (0.25–33); circularity (<0.602); FAK signal to noise ratio (>2.55), FAK integrated density (SKMEL147 > 4200; WM88 >9400), paxillin signal to noise ratio (>4.5),

paxillin integrated density (>8000), presence of F-actin (phalloidin) staining; and limited proximity to DAPI staining (integrated density <4000).

## Protein–protein interaction via mammalian 2-hybrid system

HEK293T cells were seeded at 400,000 cells per well in six-well plates in complete DMEM and allowed to adhere overnight. Once the cells have reached approximately 50–70% confluency, they were washed with DPBS and pretreated with a selection of UA-2012 dose concentrations for 2 h: 100, 50, 25, 10, and 1 µM. After the 2 h pretreatment, the Lipofectamine 2000 transfection: vector mixtures were added directly into the treated media. Cells being treated with peptide or DMSO-vehicle only received all four plasmids: 1 µg of pM-Paxillin, 1 µg of pVP-FAK-CD, 1 µg of pGL4, and 0.2 µg of pRL-SV40. An untransfected control group was included, where cells were neither treated nor transfected but allowed to proliferate for the duration of the experiment. Single-construct controls were also created with cells that did not receive treatment. The pM-Paxillin control received 1 µg of pM-Paxillin, 1 µg of pGL4, and 0.2 µg of pRL-SV40 only. The pVP-FAK-CD control received 1 µg of pVP-FAK-CD, 1 µg of pGL4, and 0.2 µg of pRL-SV40 only. Transfection in serum-free media was allowed to incubate for the next 24 h. Then cells were lysed using luciferase lysis buffer with added 1:500 DDT for 30 min and centrifuged at max speed for 30 min. Equal amounts of lysate and Dual-Glo luciferase reagent were added to white-walled, white bottom 384-well plates and allowed to incubate at room temperature, shaking for 10 min. BMG LabTech ClarioStar Plus plate reader using BMG ClarioStar MARS software (version 4.20) obtain luciferase readings via luminescence. Stop & Glo reagent is added to each well (the same volume as lysate added previously) to quench the firefly signal and provide renilla luciferase values.

## KINOMEscan™ profiling

This experiment was performed by Eurofins KINOMEscan™ Profiling Service. In brief, kinase-tagged T7 phage strains were grown in an *E. coli* host derived from the BL21 strain to log-phase and infected with T7 phage from a frozen stock (multiplicity of infection = 0.4) and incubated with shaking at 32 °C until lysis (90–150 min). The lysates were centrifuged and filtered, the remaining kinases were produced in HEK-293 cells and subsequently tagged with DNA for qPCR detection. Streptavidin-coated magnetic beads were treated with biotinylated small molecule ligands for 30 min at room temperature to generate affinity resins for kinase assays. The liganded beads were blocked with excess biotin and washed with blocking buffer (SeaBlock (Pierce), 1% BSA, 0.05% Tween-20, 1 mM DTT). Binding reactions were assembled by combining kinases, liganded affinity beads, and test compounds in 1x binding buffer (20% SeaBlock, 0.17x PBS, 0.05% Tween-20, 6 mM DTT). Test compounds were prepared as 100x stocks in 100% DMSO and directly diluted into the assay. All reactions were performed in polypropylene 384-well plates in a final volume of 0.02 ml. The assay plates were incubated at room temperature with shaking for 1 h and the affinity beads were washed with wash buffer (1x PBS, 0.05% Tween-20). The beads were then resuspended in elution buffer (1x PBS, 0.05% Tween-20, 0.5 µM non-biotinylated affinity ligand) and incubated at room temperature with shaking for 30 min. The kinase concentration in the eluates was measured by qPCR.

## Annexin V apoptosis assay

SK-MEL-147 cells were seeded at 1000 cells/well in 384-well white-walled, clear-bottom plates (Corning). Cells were allowed to adhere overnight. Serial twofold dilutions of drugs were prepared starting at 200 µM. Immediately after adding the drug treatments, 2X RealTime-Glo™ Annexin V Apoptosis detection reagent (Promega) was added. Replicates were also included for the following controls: "no-cell" (negative), vehicle-only controls (negative), and cisplatin (5 µM, positive). The BMG LabTech ClarioStar Plus plate reader using BMG

ClarioStar MARS software (version 4.20) was used to detect luminescence signals caused by Annexin V reagent binding. Dose-response curves and $EC_{50}$ curves were obtained using GraphPad Prism (version 10.3.1), reporting mean luminescence relative to DMSO ± SD. A minimum of 5 independent biological runs ($n = 5$) were performed per treatment group.

## Transwell invasion assay

SK-MEL-147 cells were seeded at 400,000 cells/well in six-well plates and allowed to adhere in the incubator overnight. Cells were pretreated with drugs ranging from 200–0.01 µM, and incubated for 48 h. Subsequently, 8.0 µm PET membrane cell culture inserts (Falcon) were coated with 40 µL of 0.3 mg/mL Matrigel (Corning) diluted in serum-free media and incubated at 37 °C/5% $CO_2$ for 5 h to allow the solution to gel. Unbound Matrigel was removed from the inserts, and membranes were rehydrated with serum-containing media for an additional hour. Treated cells were then trypsinized, collected by centrifugation, washed with DPBS, and resuspended in serum-free media. About 50,000 cells per treatment group were seeded into upper chambers with additional serum-free media. 10% FBS-containing media was placed into bottom chambers as a chemoattractant. After 24 h, inserts were then stained with crystal violet (Sigma-Aldrich) and washed with DPBS. Using an ECHO Revolve microscope, bright field images of 6 fields of view for each insert were taken at 10x magnification. The number of invaded cells in each image was quantified using ImageJ (version 1.52a). Bar graphs, dose-response curves, and $IC_{50}$ values were established using GraphPad Prism (version 10.3.1), reporting the mean number of invaded cells relative to DMSO ± SD.

## DMPK

Cell permeability of peptide drugs was determined by fluorescence measurement of TAMRA-labeled peptides in 96-well plates and ClarioStar Plus plate reader using BMG ClarioStar MARS software (version 4.20). Drugs were allowed to permeate cells for 48 h, followed by PBS washing. Percent uptake was calculated as background-subtracted fluorescence absorbed by cells/fluorescence of peptide in media-only wells, then made relative to TAMRA-Alkyne. Three biological replicates ($n = 3$) were performed per cell line per treatment group. Peptide proteolysis was measured using immobilized Trypsin-agarose resin (Thermo Fisher) and quantification by LC-MS. In brief, calibration curves for peptides were determined by HPLC quantification (UV 214 nm). About 100 µM of peptide was incubated with 5 µL of trypsin resin in 0.1 M $NH_4HCO_3$ buffer for multiple timepoints (0, 0.25, 0.5, 1, 1.5, 2, 3, 4, 6, 8, 24, and 48 h). After centrifugation, peptide supernatants were quantified by HPLC, and half-life was calculated by GraphPad Prism (version 10.3.1). In vivo pharmacokinetics were performed by the UACC Analytical Chemistry Shared Resource. All procedures were approved by the University of Arizona Institutional Animal Care and Use Committee (IACUC). A single intraperitoneal (i.p.) administration of peptide 2012 dissolved in peptide diluent (85% saline, 10% solutol, 5% DMA) was given to 8-week female Balb/cJ mice (Jackson Laboratories, Bar Harbor, ME) ($n = 3$ per timepoint) and blood was collected by retro-orbital collection at various timepoints (0, 0.5, 1, 2, 4, 8, and 24 h). The mice were housed in ventilator racks (Lab Products, LLC Seaford, DE), maintained under specific pathogen-free conditions, with controls to maintain 20–24 °C and 30–70% humidity, and on a 12 h light/dark schedule. The mice were fed NIH-31 irradiated pellets (Tekland Premier, Madison, Wisconsin), and reverse osmosis water was freely available. Sentinel mice were screened monthly by ELISA serology for mycoplasma, mouse hepatitis virus, pinworms, and Sendai virus and tested negative. Plasma concentrations of peptide 2012 were determined on a TSQ Quantum Ultra LC/MS system. The average plasma concentration versus time data were analyzed using PKSolver (version 2.0) by a noncompartmental approach.

## In vivo melanoma model

The B16F10 mouse melanoma model was performed in collaboration with the UACC Experimental Mouse Shared Resource. All procedures were approved by the University of Arizona Institutional Animal Care and Use Committee (IACUC). In brief, 8-week female C57Bl/6J mice (Jackson Laboratories, Bar Harbor, ME) were housed in ventilator racks (Lab Products, LLC Seaford, DE), maintained under specific pathogen-free conditions, with controls to maintain 20–24 °C and 30–70% humidity, and on 12 h light/dark schedule. The mice were fed NIH-31 irradiated pellets (Tekland Premier, Madison, Wisconsin), and reverse osmosis water was freely available. Sentinel mice were screened monthly by ELISA serology for mycoplasma, mouse hepatitis virus, pinworms, and Sendai virus and tested negative. Mice were injected with $5 \times 10^4$ B16F10 melanoma cells (ATCC, Cat #CRL-6475) into the flank of the mouse using a subcutaneous (s.q.) injection. Once tumors were palpable (50–100 $mm^3$), mice were randomized into treatment and control groups ($n = 8$ per group). Peptide was administered i.p. at 20 mg/kg q.d. for a total of 20 days. Peptide diluent (85% saline, 10% solutol, 5% DMA) was used as a control. Tumor volume ($mm^3$) was measured using calipers and calculated using the following Eq. 4:

$$V = \frac{a^2 \times b}{2} \tag{4}$$

where a is the smallest diameter and b is the largest diameter. Mice weight was measured three times per week. Mice were humanely euthanized upon endpoint of large tumor burden (>2000 $mm^3$) or if mice became moribund. Necropsies were performed to examine gross toxicity.

## Reporting summary

Further information on research design is available in the Nature Portfolio Reporting Summary linked to this article.

## Data availability

All data generated or analyzed during this study are included in this published article and its supplementary information files. X-ray crystallographic data generated from this report is available in the Protein Data Bank (PDB) repository under accession code 6PW8. Accession codes of previously published data used in this work are 1OW8 and BMRB 28012. Source data are provided as a Source Data file. Source data are provided with this paper.

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

## Acknowledgements

We would like to thank Drs. Michele Zacks and Nirupa Nagaratnam for help in the review and editing of the manuscript. We would like to acknowledge the Magnetic Resonance Research Center (Dr. Brian Cherry) at Arizona State University for help in the design & collection of HSQC-NMR data. We would like to acknowledge the University of Arizona Cancer Center Experiment Mouse Shared Resource (EMSR) and Gillian D. Paine-Murrieta for help in the design and collection of data for animal experiments. We acknowledge the Arizona State University Biodesign Center for Applied Structural Discovery for support of the structural studies. Research reported in this publication was supported by the National Cancer Institute of the National Institutes of Health under award numbers R41CA240124, R42CA240124, R37CA065910, and P30CA023074; by the National Institute of General Medical Sciences of the National Institutes of Health under award number DP2GM132931; and by the National Science Foundation under midscale RI-2 award No. 2153503.

## Author contributions

L.R., L.N., W.S.W., D.T., E.S., L.M., R.N., C.Q., A.B., C.A., A.R., J.A.M., R.K. R.N., B.R.C., J.M.M.-G., and T.M. acquired and interpreted data; N.S., R.F., P.F., W.C., and T.M. supervised studies; L.R., L.N., W.S.W., D.T., E.S., B.R.C., R.F., P.F., and T.M. contributed to manuscript writing; all authors revised and approved the final manuscript.

## Competing interests

Timothy Marlowe, Ph.D. is Chief Scientific Officer and has an equity interest in FAKnostics LLC, which has been disclosed to the University of Arizona. William Cance, M.D. has a family member with an equity interest in FAKnostics LLC. Conflicts of interest resulting from this interest are being managed by The University of Arizona in accordance with its conflict-of-interest policies. The remaining authors declare no competing interests.
