## [Transparent Peer Review file · Nature Communications]

Structure-based discovery of hydrocarbon-stapled paxillin peptides that block FAK scaffolding in cancer

Corresponding Author: Dr Timothy Marlowe

Version 0:

Reviewer comments:

Reviewer #1

(Remarks to the Author)

In this work, the authors describe a peptide that acts as a paxillin LD-motif mimetic that they propose to inhibit the paxillin interaction with focal adhesion-targeting (FAT) sequences in FAK (although I note this is not shown in cells). They generated so-called stapled/cyclised variants, with evidence of two site binding from NMR studies, and biochemical analyses are proposed to have greater binding affinity than the LD2 sequence which it was based on. They also had greater stability in the presence of trypsin and enhanced cell permeability after addition of a myristylation group. They additionally solved and reported the co-crystal structure of the FAT:peptide (called 1907) complex. This peptide chemistry part of the paper all looks solid – although this is not my area of expertise and I cannot comment on the validity of conclusions in detail.

I agree with the authors that agents that would efficiently disrupt FAK's protein-protein interaction or scaffold functions may be more effective as an intervention approach than inhibiting FAK's relatively weak kinase. Therefore, with the caveats mentioned about my lack of expertise in assessing, the paxillin:FAK-FAT disruptive agent they have looks like an interesting compound to investigate. However, I think the biological investigations so far (Figure 5) are too preliminary and further in-depth analyses are required.

Specific points on the biological effects in Fig 5:

1. Only a single melanoma cell line is shown. How do the authors know these are focal adhesions (FAs) (5d), besides staining with FAK. Could they stain for other conventional FA proteins, including paxillin?
2. How does the fluorescence tag to assess cell permeability affect the drugs ability to out-compete the FAK-paxillin interaction? (is this believed to be competition in cells?)
3. What happens to paxillin when the putative paxillin-FAT disrupting peptides are present? Do focal adhesions not form or is FAK just missing from them? What happens if the drug is washed out? Do cells recover? It is weird that the cells are still attached and spread when they are also undergoing apoptosis – which generally causes them to detach. Can the authors comment on timing and doses of the treatment and when various strands of the biological effects become evident – removal of FAK from FAs and apoptosis.
4. 1uM is quite a high concentration – some pharmacological dose response and time of treatment curves of biological effects AND, importantly, effects on in-cell FAK-paxillin complex would be very helpful. There needs to be some evidence to imply the biological mechanism of action is as proposed. Can the authors rule out a general toxicity that is unrelated to FAK removal from focal adhesions.
5. The tumour versus normal proposed specificity point is too preliminary for such a conclusion. This needs more work/more cell lines to be examined.
6. I wonder how to interpret the in vivo experiment, which could be interesting. Can the authors think of a way to test whether the observed beneficial effect is linked to the proposed interference with FAK-paxillin complex in cells within the tumour. I doubt the tumours in vivo have substantial focal adhesions as we define these in cells in culture; hence, important to know whether the removal of FAK from adhesions is still the relevant mechanism in vivo.

Reviewer #2

(Remarks to the Author)

Focal adhesion kinase (FAK) is a multidomain non-receptor tyrosine kinase. Due to its importance for cancer cell survival and invasiveness, FAK is recognized as an important target for anti-cancer therapies. Many cellular effects of FAK in normal and cancerous cells stem from its scaffolding function. This dual function of FAK as a kinase and scaffold may contribute the limited anti-cancer effect of small molecule inhibitors against the FAK kinase activity.

Naser et al. aimed at designing hydrocarbon stapled peptides capable of inhibiting FAK by binding to its C-terminal focal adhesion targeting domain (FAT). These peptide candidates were designed to mimic so-called LD motifs with which paxillin interacts with FAT to recruit FAK to focal adhesions. From the total of 19 peptide candidates that the authors designed and tested, they identified a stapled peptide, termed 1907, that binds to both available LD motif binding sites on FAT. The binding affinity for one of the two FAT sites was 100-fold greater than the natural paxillin LD2 motif, making this stapled peptide a good competitor for paxillin binding. The high affinity binding of this peptide to the relatively flat LD interaction surface of FAT is demonstrated by NMR and the 1.95 Å crystal structure of the 1907:FAT complex.

Based on this finding, the authors modified 1907 by adding a myristoyl (resulting in what they call peptide 2012).

Myristoylation markedly improved cell permeability. 2012 was able to relocate cellular FAK away from focal adhesions, promoting anti-cancer effects that the authors claim superior to the more traditional approach of inhibiting the FAK kinase domain. The authors suggest that their approach can also be used for targeting other protein-protein interactions surfaces that would be difficult to inhibit with small molecule compounds.

Overall, the work is well described. Its claims are justified by a large array of methods for in vitro binding and characterization (SPR, FP, CD), as well as structural (NMR, X-ray) and functional (in vivo pharmacokinetics, studies in cells and tumour mouse model) evaluation.

Minor comments:

Hydrocarbon stapled peptides have already been used for about a decade by several groups to target protein-protein interactions (e.g. Verdine & Hilinski 2012, Ali et al. *Comp. Struct. Biotechnol J.* 2019; Lai et al. *Nature* 2022), some of which have entered clinical trials (Bluntzer et al, *Peptide Sci.*, 2020). Also the use of myristoylated hydrocarbon stapled peptides has been reported previously (Cromm, *ChemBioChem* 2019). Hence, the conceptual peptide design is not novel as the Abstract appears to claim. However, its successful application for the important anti-cancer target FAK may inspire novel drugs for this specific protein. Therefore, I would suggest rephrasing the last sentence of the Abstract to better reflect the specific advance achieved.

Figure 2: for the inhibition assays, it would be good to state in the legend the concentration used of LD2; and that it was a competition with their TAMRA-LD2 L10D peptide.

Supp Fig 8: the hydrocarbon staple electron density should be an omit density (or polder map). Here it is unclear what it is, but presumably a 2FoFc.

Binding of 1907 was only submicromolar on one site, whereas paxillin can bind with micromolar affinity to both sites, due to its multiple binding sites. This raises the question whether the affinity of the single-site 2012 is strong enough to compete against multivalent paxillin? In this context, Supp Fig 10 is quite important to support the author's claims and I would suggest putting it into the main figures.

Line 299: "indicating potential [defactinib] toxicity beyond the 2.5 µM". Is there literature evidence for such a sharp increase in toxicity for this compound?

Line 356: "FAT domain of FAK has limited sequence similarity". I would guess that the PYK2 FAT domain is also affected by the peptide 2012? Could the authors speculate about a possible benefits/effects of a potential dual inhibition?

Reviewer #3

(Remarks to the Author)

Naser et al have submitted a high quality manuscript describing the discovery and validation of a hydrocarbon stapled peptide that inhibits the association of FAK and paxillin in a manner distinct from kinase domain inhibition. The authors conduct excellent biophysical and biochemical experiments that yield a conclusive in vitro data package. However, the cellular results are underdeveloped. The manuscript would benefit from buttressing these modest deficiencies.

- Please include 2012 in Supp Table 1 and provide an LCMS trace of 2012.
 - Please confirm that FBS was indeed absent for the viability assays. Methods imply that cells were washed with PBS to remove serum before treatment. If true, repeat the cellular assays with at least 1% FBS, preferably, 5% (or add back 10% FBS after 2 hrs of treatment), as viability data for many days of serum withdrawal is not physiologically relevant.
 - Non specific membrane lysis: the lipidated peptide could be membrane disruptive—confirm with LDH release in several cell lines at high dose... at least 25 µM
 - Verify therapeutic window in multiple additional cell lines, both responsive and non-responsive. Was the compound really not tested in the B16-F10 cell line before the in vivo experiment?
 - The intracellular target binding experiment is a critical component of this manuscript. Conduct this immunofluorescence experiment across multiple additional doses and cell lines. Further, confirm that peptide is responsible for this, as in co-incidence of TAMRA-peptide and green foci. Alternatively, make a biotin labeled peptide and confirm target pulldown in cells or co-IP disruption of foci with peptide treatment.
- o Supp Fig 9 appears designed to address these points, but a few omissions prevent this. Please include a TAMRA 2020.

How many cells were sampled for these? How are green levels normalized? Several foci would be more apparent in 2012 treatment if the green level was higher. Also please omit polynucleated cells from the analysis.

o Similarly with Sup Fig 10, more conclusive results would be attained with a fuller dose response for 2012, so as to make a better bridge the gap between DMSO and 1 μM . Also please include the 2020 negative control. Lastly be sure to include details for the M2H method.

- How many mice for in vivo? Methods state $n=8$, while legend implies only $n=3$ for some arms. Toxicity needs to be studied in another model, as B16-F10 model is known not to lose weight.
- The authors need to investigate or reconcile the discrepancy between the modest cellular activity and rather compelling in vivo activity ($\sim 2 \mu\text{M}$ cellular IC_{50} , $\sim 20 \mu\text{M}$ IC_{90} in cells). Furthermore, what was the C_{max} from the PK study?
- The clever idea of appending myristic acid shows a dramatic improvement and cell uptake and efficacy, but further context is needed. For instance, does the drug dissipate into every membrane throughout the entire volume of the mouse or is it extensively serum bound?

Reviewer #4

(Remarks to the Author)

The manuscript by Naser, Weiner, Thifault et al. investigates whether a novel FAK-scaffold inhibitor, a hydrocarbon stapled peptide, may be a potential future therapeutic in the treatment of various cancers. The authors characterize the interaction between a stapled peptide derived from the paxillin LD2 alpha-helical motif and the FAT domain of focal adhesion kinase using x-ray crystallography and heteronuclear single quantum coherence-nuclear magnetic resonance. They show that myristoylation improves cell permeability and the metabolism and pharmacokinetic properties of the stapled peptide, which appears to act as a FAK-scaffold inhibitor that can induce apoptosis and reduce cell invasion and tumor burden in a mouse model of melanoma. Although these results are very interesting and quite promising, there are several issues the authors need to address. These are discussed below:

1. The identification of focal adhesions using FAK antibodies is not convincing. Colocalization with another component of focal adhesions, such as paxillin, vinculin, or talin, along with the observed punctate staining would be more convincing. Although this comment applies to both Figure 5d and Supplementary Fig. 9, it is especially true for Supplementary Fig. 9 where red arrows are allegedly pointing to focal adhesions identified using anti-FAK alone. However, these alleged focal adhesions are very difficult, if not impossible, to see, thus making their presence questionable. This issue also makes the quantitation that is provided very difficult to assess.
2. Given that patients with melanoma don't die from their primary tumor(s) and that this peptide is being promoted as a potential future therapeutic, it would be more clinically relevant to also examine the mouse lungs or other tissues to determine whether the peptide either prevented metastasis or reduced metastatic tumor growth once metastasis had occurred. In the Methods section, the authors mention that a gross necropsy was performed to examine the presence or absence of metastasis. Were metastases quantitated or were metastatic tissues collected for histology and quantitation? Such data would significantly enhance the importance of this work.
3. The authors do not reference the highly relevant Mousson et al. paper (Cancers, 2021, 13, 1871). The authors should consider referencing this paper because it provides further support for their approach.
4. Myristoylated hydrocarbon-stapled 2012 is far more stable than the native peptide 1907. What about the control peptides 2014 and 2020? Peptide 2020 is used as a control in the in vitro cellular assays and knowing its stability profile would help in evaluating its relevance as a good peptide control. In this same regard, is peptide 2020 cell permeable?
5. The functional in vitro studies were performed on SK-MEL-147 melanoma cells. To show broader applicability to other cancers it would be helpful to have these assays performed on another cancer cell line, such as the SK-BR-3 breast cancer cell line the investigators used for other assays in the manuscript.
6. The authors do not describe what measurements were taken to determine tumor size in their in vivo studies nor do they indicate how these measurements were used to calculate tumor volume. A common, but potentially inaccurate, method is to measure the length and width of the tumor with calipers and then assume that the width and depth of the tumor are equal (which can lead to large errors in tumor volume estimates) and that π is equal to 3 (a reasonable approximation). By making these assumptions, a simplified version of the formula for an ellipsoid can be derived, and this formula ($\text{Volume} = \text{length} * \text{width}^2/2$) is often used in the literature but can lead to inaccurate volume estimates as noted above. When all three dimensions of the tumor are measured, the full formula of an ellipsoid can be used to generate more accurate volume estimates. Regardless of what method was used, the authors need to provide this information in their Methods section.
7. One-way ANOVA was used in several cases when comparing results between multiple groups or treatments, and results of these analyses were provided as a statistically significant p values—meaning that the overall analysis of data from all groups provided a statistically significant result. However, it does not appear that multiple comparisons tests were performed, and yet asterisks were placed over specific groups, suggesting that these particular groups/treatments were statistically significant when compared to another group or set of groups. Furthermore, in the legend for Supplementary Fig. 9, the authors specifically show/state that their data generated a p value < 0.0001 for both 1 μM 2012 vs DMSO and 1 μM 2036 vs DMSO, implying that they did a multiple comparison test. If so, what multiple comparisons test(s) was/were used to

make these assessments?

In Supplementary Fig. 9, there appears to be no variation (error bars) in the DMSO control. Does this reflect the actual data or were values in each independent experiment normalized to that experiment's control, thus making all values in the control equal to 1? If the latter explanation is true, a normal one-way ANOVA that includes the control would be inappropriate because all variability in the control would have been eliminated during the normalization process. If so, the authors should consider not normalizing their data or normalizing in a manner that will retain the variability in their control. Otherwise, they may have to log transform their data and use a one-way randomized block ANOVA (same as repeated measures in Prism) with an appropriate multiple comparisons test. Consulting a statistician may be helpful in this case.

8. Two-way ANOVA was performed for the data shown in Figure 5j. Again, a multiple comparison test would need to be performed to determine whether individual time points are significant. What multiple comparison test was performed to determine that data for days 22, 26, and 29 were statistically significant?

9. The Discussion could be shortened somewhat by eliminating unnecessary reiteration of the approach and results in the second and possibly the third paragraph. A short review of the results to put the discussion in context is valuable, but it seems that there is too much reiteration in these two paragraphs.

Minor points:

1. On line 443, it appears the word "described" is missing as in "previously described."

2. On line 483, "E.coli" should be italicized "E. coli."

3. On line 485, the first word on the line should be 1-thiogalactopyranoside.

4. In the paragraph encompassing lines 489-497, the three references to "gluthione" should all be "glutathione."

5. On line 530, it seems that the word "was" should be "were."

6. Given that the investigators potentially have a negative control peptide 2020, why was it not used in the in vivo studies? Was this a matter of expense? If so, it is understandable, although comparing 2012 to an available similar negative control peptide rather than the vehicle would have been preferred.

7. The authors should consider showing all data points instead of bar graphs or showing a combination of both. This allows readers to get a better sense of the actual data.

8. When comparing panels f and g in Fig. 5, it appears that defactinib reduces cell viability through some mechanism other than apoptosis. Is this the authors' interpretation of the data as well? In panel g (Annexin V assay), defactinib-treated SK-Mel-147 cells exhibit a concentration-dependent decrease in luminescence. Is this because the cells are dead or dying by some mechanism other than apoptosis? Was necrosis also assayed? If so, what was the result? Although defactinib is not the focal point of this study, it is used as a control, and therefore, understanding its mode of action compared to peptide 2012 is relevant.

9. According to the supporting documents, only female mice were evaluated in the reported in vivo studies. This should be noted in the "In vivo melanoma model" section of the Methods. A short note as to why only female mice were used in the study may also be appropriate.

Reviewer #5

(Remarks to the Author)

The interaction between FAK's focal adhesion targeting (FAT) domain and paxillin is important to modulate various aspects of cancer. Herein, the authors report on the structure-based optimization and synthesis of hydrocarbon-stapled peptides targeting the FAK FAT/paxillin complex. In details, through a variety of biophysical studies (including interaction assays by FPA, SPR and NMR techniques), they describe the discovery of a stapled peptide (1907), that mimics the paxillin LD-motif, and can block FAT/paxillin interaction. Peptide 1907 presents a 100-fold increase in binding to the FAT domain with respect to the native LD2-sequence and represents a dual-site binder of the FAT four-helix bundle structural organization. The structure of the 1907 peptide/FAT complex was determined as well by X-ray crystallography. Such kind of peptides generate better anti-cancer effects than the already reported FAK-kinase domain inhibitors. Indeed, a myristoylated peptide 1907 (=peptide 2012) can induce delocalization of FAK from focal adhesions, favors cancer cell apoptosis, decreases in vitro viability and invasion, and it is also able to lower tumor burden in B16F10 melanoma mouse models.

The manuscript is interesting and well written, but the following points need to be clarified.

1- NMR studies in Fig.3. The authors run the chemical shift perturbation experiment by employing FAT domain at 100 micromolar concentration and the peptide at the following concentrations: 5, 10, 50 micromolar (and lower...). Is it not clear if the "saturation condition" was reached: the authors should run the same HSQC experiment at higher peptide concentrations, let's say between 75 uM and 100 uM, and demonstrate that the resulting HSQC spectra are identical (=no more changes are occurring) respect to what observed at 50 micromolar. This is a prominent issue as Figure 3 A is not clear: a better larger image might help readers to better understand the data. Moreover, such NMR studies are usually analyzed by reporting in graph (histograms) the chemical shifts and/or intensity deviations of the apo FAT domain (first control HSQC with no peptide

and DMSO) respect to completely peptide bound FAT domain (=saturation point) for each single residue over the whole FAT sequence. The authors should produce such graphs, choose a threshold for chemical shifts and/or intensity variations and color consequently the residues most affected by binding on the 3D protein structure. Chemical shifts variations should be averaged values keeping into accounts ¹⁵N and ¹H deviations (Please see for example Morin G, Fradet-Turcotte A, Di Lello P, Bergeron-Labrecque F, Omichinski JG, Archambault J. A conserved amphipathic helix in the N-terminal regulatory region of the papillomavirus E1 helicase is required for efficient viral DNA replication. *J Virol.* 2011 Jun;85(11):5287-300. doi: 10.1128/JVI.01829-10. Epub 2011 Mar 30. PMID: 21450828; PMCID: PMC3094988., Di Lello P, Miller Jenkins LM, Mas C, Langlois C, Malitskaya E, Fradet-Turcotte A, Archambault J, Legault P, Omichinski JG. p53 and TFII α share a common binding site on the Tfb1/p62 subunit of TFIIH. *Proc Natl Acad Sci U S A.* 2008 Jan 8;105(1):106-11. doi: 10.1073/pnas.0707892105. Epub 2007 Dec 26. PMID: 18160537; PMCID: PMC2224167; Molecular Recognition of Paxillin LD Motifs by the Focal Adhesion Targeting Domain Maria K. Hoellerer, Martin E.M. Noble, Gilles Labesse, Iain D. Campbell, Jörn M. Werner, Stefan T. Arold
<https://doi.org/10.1016/j.str.2003.08.010>

.... but there are many NMR related manuscripts showing such kind of analyses).

Figure 3 panel b is not clear: the peaks should be reported as contour lines and not with solid filling. Panel c: it is very difficult to get quantitative KD values by NMR especially under strong binding / slow exchange conditions, so what equation did the authors employ to determine KD values (See also work by Farmer (Farmer BT 2nd, Constantine KL, Goldfarb V, Friedrichs MS, Wittekind M, Yanchunas J Jr. Robertson JG, Mueller L. *Nat. Struct. Biol.* 1996; 3:995–997.))? The employed equation and a proper reference should be introduced and nevertheless, it is again important to know if the last point of titration corresponds to the saturation point to get a reliable kd estimate.

In summary the NMR study appears very qualitative and better / quantitative interpretation of data -as suggested above- is necessary.

Graphs of chemical shifts and/or intensity changes vs residue numbers need to be introduced; nevertheless, an overlay of HSQC spectra at two high concentrations of peptides for example 50 μ M and 75-100 μ M is necessary to demonstrate that saturation conditions were reached at 50 μ M peptide concentration.

Moreover, the authors compared the HSQC spectrum of protein with 1% DMSO only with HSQC spectra of protein/peptides - with peptide at diverse concentrations-. Does 1% DMSO correspond to the condition 50 micromolar peptide or was it kept constant at all peptide points?

2- SPR suggests that the peptide 1907 is a dual site binder able to target helices 1-4 and 2-3 in FAK FAT domain. Did the authors try to fit SPR data with a single-site binding model? Is this possible? Why is only a single KD SPR value indicated for the LD2 pep in table 1?

It is strange that X-ray can see only electron density at the 2-3 site Could the NMR shifts -observed at site 1-4- be related to conformational movements rather than direct binding? I mean small conformational movements of the protein after binding of the peptide only at the 2-3 site (?). This point can be eventually better addressed with some kind of displacement experiment by employing a ligand specific for the 1-4 single site (if it is available) and an FPA assay. ITC measurements between FAT/1907 could also give additional information on the binding stoichiometry.

3-CD data, Figure S5 the author say "...demonstrate that hydrocarbon stapling can increase α -helical content by 10 % in aqueous sodium phosphate solution (pH 7.0)". They should clearly indicate how they calculated the 10% increase.

4-Minor: in the methods 16 scans per "FIP" ...should be "FID"

Version 1:

Reviewer comments:

Reviewer #1

(Remarks to the Author)

I have only re-read the biological consequences data. I think this manuscript has been improved by the authors. The points I raised have been somewhat addressed and the case is stronger. There is a couple of bits of information I requested that were not referred to in the rebuttal letter I don't think, for example - point 3 - What happens if the stapled peptides are washed out? Do cells recover? I am sure the authors know the answer to this and could perhaps comment. I also think the time-dependent effects may have been part of the first description/basic characterisation of effects, rather than some follow up experiments for next mechanistic studies as mentioned.

Reviewer #2

(Remarks to the Author)

The authors have responded in a satisfactory manner to my comments

Reviewer #3

(Remarks to the Author)

Well done, no further comments

Reviewer #4

(Remarks to the Author)

The authors have addressed the points I raised in my original review, and I only have a few remaining comments and suggestions, all of them relatively minor, except for the first cautionary note.

1. As I suggested the authors do in the original review, they have now included the formula used for estimating tumor volume. The one they have chosen to use is the less accurate but commonly used formula for calculating tumor volume, which is derived (see below) from the formula for the volume of an ellipsoid. This simplified formula can lead to large errors in estimated tumor volume if the width diameter (called the smallest diameter in the authors' response) and depth diameters (not measured in this study but assumed to be the same as the width diameter) of the tumor are substantially different. Regardless, this derived formula is commonly employed in tumor studies and the literature is replete with its use. In many cases, using this formula might not change the conclusions of the experiment because all tumor volumes are calculated using the same formula. However, a problem in interpretation could arise if the unmeasured depth diameters of different tumors were substantially different from each other. To better understand this point, the formula for the volume of an ellipsoid and the derivation of the simplified formula are shown below:

Ellipsoid volume = $\frac{4}{3} \pi \cdot \text{length}/2 \cdot \text{width}/2 \cdot \text{depth}/2$

If one assumes that $\pi = 3$ (a reasonable approximation) and the width and depth of the tumor are the same, this formula simplifies to the formula used in the paper, which is $\text{width}^2 \cdot \text{length}/2$ or $a^2 \times b/2$.

One can readily see that if the depth of two tumors is substantially different, but their length and width are the same, the derived formula will lead to the two tumors being estimated as being the same size when, in reality, their volumes could differ by several-fold. Therefore, caution should be exercised when using the simplified formula.

2. Minor issues and typos that should be addressed:

a. On line 898, "Dunnet's" should be "Dunnett's"

b. In Supplementary Figures 12 and 14, I think 1.25 μM should be 12.5 μM

c. In the Supplementary Data document, the authors should consider making their figure legends consistent with each other and the actual figures by using all lower-case letters (possibly bold font) for panel labels (see Supplementary Figure 15 where uppercase letters were used) and by enclosing the panel labels in parentheses. Other methods can be used but consistency looks good.

d. Point of clarification for a discrepancy. The manuscript (which is the important document) states that Sidak's multiple comparisons test was used for data shown in Figure 6i while the rebuttal letter (less important) indicates that Dunnett's multiple comparison test was used. Please clarify.

Reviewer #5

(Remarks to the Author)

I appreciated the authors performed more experiments and put efforts in replying to my comments, however, they still need to revise the manuscript to further clarify the NMR data as not all my queries have been properly addressed.

1)-I asked if saturation conditions were reached and the authors replied "Saturation is dependent on the residue. As seen in Supplemental Figure 6, which is a contour plot of specific residues, saturation is achieved at various concentrations. For residues in the 2-3 helices, we see it at as low as 10 micromolar. However, for residues at the 1-4 helices, those residues appear to be saturated at 100 micromolar."

Well, saturation means that all binding site positions are occupied, and no more changes are observed in the HSQC spectrum of the protein upon further addition of the peptide. So, in theory from what the authors are saying it is still not clear if saturation was reached. Are they sure that if they increase the concentration of peptide, let's say from 100 micromolar to 200 micromolar, no more changes in the protein HSQC spectrum will occur? Can they show another figure with overlays of HSQCs of protein alone (DMSO control); protein + peptide at 6 micromolar; protein + peptide at 100 micromolar, and protein + peptide at higher concentration -let's say 200 micromolar-? I really would like to see if the two HSQCs collected at 100 and 200 micromolar peptide are identical (or almost identical)...this would prove saturation is reached.

I'm not surprised that different residues behave differently, this always happens in this case as there are two sites affected by binding, the residues involved the most in peptide binding will likely saturate early. ...but the complete saturation will be reached only after you won't observe further changes in any peak of the HSQC spectrum upon further addition of peptide. KD evaluation by NMR is more accurate if saturation conditions are reached. If you are not sure if saturation was reached a comment in the text should be added.

2)-Although the authors claim they run experiments at 100 micromolar peptide concentration this condition is never mentioned in the methods or other parts of the main text of the manuscript but just in Fig.S6. Please add the proper conditions in the methods NMR section.

3)-Overall, the supplementary Fig. S6 is unclear and further NMR analyses are needed.

In detail, please see Fig.3: The authors should also add to panel a in the Figure 3 the overlay with the HSQC spectrum acquired with 100 micromolar peptide; the analyses shown in panels b) and c) need to be repeated for the 100 micromolar concentration (these analyses can be added as a further Supplementary Figure).

4)-Caption of Fig.3: "Residues with 1H-15N" replace with "Residues with 1H-15N"; "incubation of FAT at with 6 μM 1907" erase "at"; "Highlighted residues match those highlighted in panel b." change the text to avoid repeating "highlighted" twice

5)-Figure 3: Panel d) the y axis label is cropped "DMS" please adjust. Overall, the Figure has low resolution. The authors are invited to improve the Figure. This Figure needs to be enlarged as unclear, the text inside the panels is too small.

6)-Regarding KD evaluation: Keep in mind that in case of slow exchange you cannot really evaluate well a KD by NMR and you should add a comment on that; please note KD can be calculated by NMR with certain equations and you should add a

reference for the equation you used please see for example : "NMR methods for the determination of protein–ligand dissociation constants Progress in Nuclear Magnetic Resonance Spectroscopy 51(2007)219–242; See also Journal of Biomolecular NMR (2022) 76:153–163 <https://doi.org/10.1007/s10858-022-00402-3>.

I have already asked clarifications about KD evaluation and the author replied: "Information about the equation that was used to calculate KD values is included in the Methods and Materials section." But in the methods I cannot see any reference to NMR studies for KD evaluation but just : "The Kd for each residue was calculated using GraphPad Prism using the One Site – Specific Binding Least Squares fit."

Version 2:

Reviewer comments:

Reviewer #5

(Remarks to the Author)

The authors replied to all the points that I raised. They improved clarity of Figure 3, they added the additional Supplementary information (See Fig.s S6-S7) and details to the method section, as I asked. I appreciated they inserted a comment about the limits of their KD evaluation by NMR and the fact that, being in slow exchange, and not having reached completely saturation, the reported Kd values by NMR can be considered only "estimates".

In my opinion the manuscript can be accepted for publication. There are minor mistakes ("typos") that I suggest to correct: 1H-15N ...1 and 15 should be superscribed and in the methods section after reporting the equation to evaluate kd, when saying " ΔV is one minus the proportion of peak volume" instead of using the word "proportion" -that sounds unusual- I would say "ratio".

Responses to Reviewer #1:

Specific points on the biological effects in Fig 5:

Comment 1:

Reviewer Comment: “Only a single melanoma cell line is shown. How do the authors know these are focal adhesions (FAs) (5d), besides staining with FAK. Could they stain for other conventional FA proteins, including paxillin?”

RESPONSE: The WM88 melanoma cell line has been added to the IF assay (Supp Fig. 13 & 14) alongside the original SK-Mel-147 cell line. Co-staining with FAK, conventional FA protein paxillin, and F-actin (phalloidin) was completed to validate punctate foci as focal adhesions present in FA complexes.

Comment 2:

Reviewer Comment: “How does the fluorescence tag to assess cell permeability affect the drugs ability to out-compete the FAK-paxillin interaction? (is this believed to be competition in cells?)”

RESPONSE: As shown in Suppl. Fig. 11, the TAMRA-tagged derivative of 2012 (2036) disrupted FAK localization to focal adhesions in SK-BR-3 cells to a similar level as unlabeled lead peptide 2012. These data suggest that the TAMRA tag itself does not block the peptide’s ability to target the FAT domain of FAK.

Comment 3:

Reviewer Comment: “What happens to paxillin when the putative paxillin-FAT disrupting peptides are present? Do focal adhesions not form or is FAK just missing from them? What happens if the drug is washed out? Do cells recover? It is weird that the cells are still attached and spread when they are also undergoing apoptosis – which generally causes them to detach. Can the authors comment on timing and doses of the treatment and when various strands of the biological effects become evident – removal of FAK from FAs and apoptosis.”

RESPONSE: We have added new immunofluorescence data (Fig. 5, Supp Fig. 12-14) in SK-Mel-147 and WM88 cells with FAK, paxillin, phalloidin (F-actin), and DAPI staining to comprehensively evaluate the effects of 2012 on the focal adhesion. Our data show that 2012 dose-dependently inhibits FAK localization ($IC_{50} = 800 \text{ nM}$) at lower doses than paxillin localization ($IC_{50} = 4.41 \text{ }\mu\text{M}$), whereas defactinib initially stabilizes FAK at focal adhesions. Negative control 2020 had minimal effects. Although it is well-established that paxillin recruits FAK to focal adhesions through FAT domain interactions (Tachibana et al., JEM, 1995), it has also been shown that FAK can co-recruit paxillin to the focal adhesion (Hu, et al., 2014), and thus it is not unexpected that higher concentrations of 2012 disrupt paxillin localization as well. We have added to the discussion section to explain these findings in further detail. Per Fig. 5 & 6, and Supp Fig. 18, our data suggest that maximal focal adhesion displacement occurs at 48h whereas most potent induction of apoptosis occurs at 96h. However, future mechanistic studies (time-dependent effects, cell death, proteomics, transcriptomics, etc.) will help supplement the findings from this initial report.

Comment 4:

Reviewer Comment: “1 μ M is quite a high concentration – some pharmacological dose response and time of treatment curves of biological effects AND, importantly, effects on in-cell FAK-paxillin complex would be very helpful. There needs to be some evidence to imply the biological mechanism of action is as proposed. Can the authors rule out a general toxicity that is unrelated to FAK removal from focal adhesions.”

RESPONSE: Our plethora of biophysical, biochemical, structural biology, and cellular data (SPR, FP, NMR, X-ray crystallography, immunofluorescence, mammalian two-hybrid) collectively indicate that the synthetic stapled peptides function through disruption of FAK from focal adhesions. We have added dose-response immunofluorescence data showing dose-dependent delocalization of FAK by peptide 2012 (IC₅₀ = 800 nM, Figure 5). LDH release assays (Supp Fig. 16) performed in human melanoma cell lines SK-MEL-147 and WM88 show no indication of general membrane toxicity of these peptides, even at high micromolar concentrations (100 μ M). Given the FAT binding affinity of the lead peptide (K_D = 0.8 μ M, Table 1, Fig. 2) and the relatively weak affinity of the native paxillin LD2 motif (K_D = 82 μ M), the relative concentration of 1 μ M is not a high concentration in our opinion. Furthermore, there is strong correlation between FAT binding affinity and cellular disruption of focal adhesion localization by peptide 2012.

Comment 5:

Reviewer Comment: “The tumour versus normal proposed specificity point is too preliminary for such a conclusion. This needs more work/more cell lines to be examined.”

RESPONSE: The cell viability assay has been expanded to include 3 human cancer cell lines and 3 normal human cell lines (Fig. 6a-d). The results show peptide 2012 does not reduce viability in normal cell lines while reducing viability in the three cancer cell lines at low micromolar doses. Conversely, defactinib inhibits cell viability in both normal and cancer cells. These data correlate with a previous report (Xu et al., JBC, 2000) showing that FAK FAT domain disruption via FRNK overexpression selectively inhibits cancer cell viability but not normal cells.

Comment 6:

Reviewer Comment: “I wonder how to interpret the in vivo experiment, which could be interesting. Can the authors think of a way to test whether the observed beneficial effect is linked to the proposed interference with FAK-paxillin complex in cells within the tumour. I doubt the tumours in vivo have substantial focal adhesions as we define these in cells in culture; hence, important to know whether the removal of FAK from adhesions is still the relevant mechanism in vivo.”

RESPONSE: Focal adhesions in vivo have been shown to be significantly important in regulating cell-ECM adhesion, cell morphology, proliferation, and migration. Albeit *in vivo* focal adhesions have been reported to be smaller in size due to the less-stiff 3D matrix (Yamaguchi & Knaut, Development, 2022), they play a similar role in cell biology. In fact, 3D cell culture experiments show that FAK and the focal adhesion complex may have an even more significant role in regulating cell proliferation and survival in 3D environments vs 2D environments (Shibue & Weinburg, PNAS, 2009; Shibue et al., Cancer Discovery, 2012).

Responses to Reviewer #2:

Comment 1:

Reviewer Comment: “Hydrocarbon stapled peptides have already been used for about a decade by several groups to target protein-protein interactions (e.g. Verdine & Hilinski 2012, Ali et al. Comp. Struct. Biotechnol J. 2019; Lai et al, Nature 2022), some of which have entered clinical trials (Bluntzer et al, Peptide Sci, 2020). Also the use of myristoylated hydrocarbon stapled peptides has been reported previously (Cromm, ChemBioChem 2019). Hence, the conceptual peptide design is not novel as the Abstract appears to claim. However, its successful application for the important anti-cancer target FAK may inspire novel drugs for this specific protein. Therefore, I would suggest rephrasing the last sentence of the Abstract to better reflect the specific advance achieved.”

RESPONSE: We thank the reviewer for this comment. The abstract has been adjusted to reflect the specific advance of this manuscript.

Comment 2:

Reviewer Comment: “Figure 2: for the inhibition assays, it would be good to state in the legend the concentration used of LD2; and that it was a competition with their TAMRA-LD2 L10D peptide.”

RESPONSE: We thank the reviewer for mentioning this clarification point. This has been added to the figure legend of Fig. 2.

Comment 3:

Reviewer Comment: “Supp Fig 8: the hydrocarbon staple electron density should be an omit density (or polder map). Here it is unclear what it is, but presumably a 2FoFc.”

RESPONSE: We thank the reviewer for this suggestion. The polder map has been added to Supp Fig. 10b.

Comment 4:

Reviewer Comment: “Binding of 1907 was only submicromolar on one site, whereas paxillin can bind with micromolar affinity to both sites, due to its multiple binding sites. This raises the question whether the affinity of the single-site 2012 is strong enough to compete against multivalent paxillin? In this context, Supp Fig 10 is quite important to support the author’s claims and I would suggest putting it into the main figures.”

RESPONSE: The data presented in Fig. 5 (FAK-paxillin immunofluorescence) and supplemental Fig. 15 (mammalian two-hybrid data), which contains full-length paxillin, support that 2012 (myristoylated analog of 1907) can out-compete full-length (i.e., multivalent) paxillin binding to FAK. In fact, the binding affinity of 2012 to paxillin is 0.8 μ M (stronger site), compared to the native interaction of 82 μ M. The observation that primary disruption of the stronger helix 2-3 site, as demonstrated by X-ray crystallography (Fig. 4), is sufficient to disrupt the full-length interaction is in alignment with previous reports (Scheswohl, J Mol Signal, 2008).

Comment 5:

Reviewer Comment: “Line 299: “indicating potential [defactinib] toxicity beyond the 2.5 μM”. Is there literature evidence for such a sharp increase in toxicity for this compound?”

RESPONSE: Previous studies have shown at 1 μM, defactinib binds to 100+ kinases, resulting in more than 65% inhibition (Cromm et al, 2018). The high non-selectivity of defactinib at low micromolar doses indicates the potential of off-target effects and toxicity beyond 2.5 μM. The Eurofins KINOMEscan data also showed that 1 μM defactinib inhibited 45 out of 97 kinases tested, whereas 2012 inhibited 0 kinases (Figure 6 and Supp. Fig. 17).

Comment 6:

Reviewer Comment: “Line 356: “FAT domain of FAK has limited sequence similarity”. I would guess that the PYK2 FAT domain is also affected by the peptide 2012? Could the authors speculate about a possible benefits/effects of a potential dual inhibition?”

RESPONSE: Future work will include further investigation on the binding selectivity of peptide 2012 and potential pharmacological effects from inhibition of FAK FAT and/or Pyk2 FAT.

Responses to Reviewer #3:

Comment 1:

Reviewer Comment: “Please include 2012 in Supp Table 1 and provide an LCMS trace of 2012.”

RESPONSE: This has been added to the supplementary information (Supp. Fig. 2).

Comment 2:

Reviewer Comment: “Please confirm that FBS was indeed absent for the viability assays. Methods imply that cells were washed with PBS to remove serum before treatment. If true, repeat the cellular assays with at least 1% FBS, preferably, 5% (or add back 10% FBS after 2 hrs of treatment), as viability data for many days of serum withdrawal is not physiologically relevant.”

RESPONSE: Cells were initially washed with PBS to remove serum before treatment, then after 24 hours, 10% FBS was added back into each well. The total assay time was 96 h, and based on cellular observations, the initial serum starvation had minimal effect on the normal health of the cells.

Comment 3:

Reviewer Comment: “Non specific membrane lysis: the lipidated peptide could be membrane disruptive—confirm with LDH release in several cell lines at high dose... at least 25 μM”

RESPONSE: LDH release assay (Supp. Fig. 16) was performed in SK-Mel-147 and WM88 cell lines after treatment with 2012 and negative control peptide 2020 at a dose range of 100-0.195 μM and showed no sign of obvious membrane disruption.

Comment 4:

Reviewer Comment: “Verify therapeutic window in multiple additional cell lines, both responsive and non-responsive. Was the compound really not tested in the B16-F10 cell line before the in vivo experiment?”

RESPONSE: The CTG assay was performed in additional cell lines: WM88, ONDA7, NHEM, HEPG2, and LX2 (Fig. 6a-d). The results indicate that at low micromolar doses, peptide 2012 is able to reduce cell viability by 50% in human cancer cell lines while showing no sign of reduced viability in normal cell lines, suggesting a large therapeutic window. Given the B16-F10 melanoma model is an established model in the literature and has been shown to be sensitive to FAT domain mutation (Kaneda et al., Cancer Letters, 2008), we did not test 2012 in vitro in the B16-F10 cell line.

Comment 5:

Reviewer Comment: “The intracellular target binding experiment is a critical component of this manuscript. Conduct this immunofluorescence experiment across multiple additional doses and cell lines. Further, confirm that peptide is responsible for this, as in co-incidence of TAMRA-peptide and green foci. Alternatively, make a biotin labeled peptide and confirm target pulldown in cells or co-IP disruption of foci with peptide treatment.

- *Supp Fig 9 appears designed to address these points, but a few omissions prevent this. Please include a TAMRA 2020. How many cells were sampled for these? How are green levels normalized? Several foci would be more apparent in 2012 treatment if the green level was higher. Also please omit polynucleated cells from the analysis.*
- *Similarly with Sup Fig 10, more conclusive results would be attained with a fuller dose response for 2012, so as to make a better bridge the gap between DMSO and 1 μ M. Also please include the 2020 negative control. Lastly be sure to include details for the M2H method.”*

RESPONSE: We have added new dose-response immunofluorescence data (Fig. 5, Supp Fig. 12, 13, and 14) showing dose-dependent delocalization of FAK (IC₅₀ = 800 nM) and FAK-paxillin interaction (IC₅₀ = 1.26 μ M) by peptide 2012. This full dose-response assay has been performed in multiple cancer cell lines, including SK-Mel-147, WM88, and SKBR3, with paxillin, FAK, phalloidin, and DAPI staining. Taking into account the new comprehensive IF data package and existing M2H data, we believe 2012 demonstrates inhibition of FAK-paxillin interaction in cells and FAK delocalization from focal adhesions. Furthermore, there is strong correlation between FAT binding affinity and cellular disruption of FAK localization to the focal adhesion, supporting on-target mechanism of action. We believe biotin-peptide studies will be supportive in future manuscripts. Regarding Supp Fig. 9 (now Supp Fig. 11), co-localization of TAMRA-2012 (2036) with FAK can be temporally challenging to resolve, since the peptide itself de-localizes FAK from focal adhesions. Nonetheless, the observed membrane staining of 2036 is suggestive of FAK displacement from focal adhesions. Given the new comprehensive evaluation of 2012 and negative control 2020 in IF experiments (Fig. 5, Supp Fig. 12-14), we do not believe additional TAMRA studies will significantly advance existing conclusions. Additional details to the IF assay, including quantification of focal adhesions, have been added to the methods section. For IF, each region of

interest (ROI) selected at random, which was approximately 2-20 cells, was captured with identical fluorophore exposure settings to ensure uniformity. Parameters were set for area, signal:noise, FAK/paxillin integrated density, and proximity to phalloidin/DAPI, to quantify focal adhesions and not background nor cytoplasmic staining. As shown in multiple reports (Hu et al. Sci Reports, 2014; Brami-Cherrier et al. EMBO, 2014), reduced FA fluorescence intensity is indicative of less FA localization. For majority of ROIs, polynucleated cells are not present; however not all can be omitted due to the goal for limitation of bias. Given the new comprehensive evaluation of 2012 and negative control 2020 in IF experiments (Fig. 5, Supp Fig. 12-14), we do not believe additional M2H studies will significantly advance existing conclusions. Additional details to the M2H assay have been added to the methods section.

Comment 6:

Reviewer Comment: “How many mice for in vivo? Methods state n=8, while legend implies only n=3 for some arms. Toxicity needs to be studied in another model, as B16-F10 model is known not to lose weight.”

RESPONSE: N=8 per group for the B16F10 melanoma mouse model. N=3 for cell biology experiments. The figure legend has been updated for increased clarity. Body weight change has been consistently used as an indirect measurement of drug toxicity (Chapman et al., Reg. Tox Pharm, 2013). The toxicity point made in the manuscript is about drug-induced toxicity, not the body weight effects of the model itself.

Comment 7:

Reviewer Comment: “The authors need to investigate or reconcile the discrepancy between the modest cellular activity and rather compelling in vivo activity (~2 μM cellular IC50, ~20 μM IC90 in cells). Furthermore, what was the Cmax from the PK study?”

RESPONSE: New dose-response immunofluorescence data (Fig. 5) indicate that the cellular potency (IC50) of 2012 is 800 nM, with full FAK displacement by 3 μM. The 2012 PK study shown in Fig. 6 shows a Cmax of 89.4 μg/mL, or 48.5 μM concentration. These data indicate a 60-fold excess of 2012 plasma concentration compared to required therapeutic levels (IC50), which we believe is sufficient to achieve substantial *in vivo* efficacy.

Comment 8:

Reviewer Comment: “The clever idea of appending myristic acid shows a dramatic improvement and cell uptake and efficacy, but further context is needed. For instance, does the drug dissipate into every membrane throughout the entire volume of the mouse or is it extensively serum bound?”

RESPONSE: Plasma protein binding (PPB) studies show that peptide 2012 has >99% binding to plasma proteins, however we view these data as beneficial for a peptide drug candidate, where high plasma protein binding can significantly extend drug half-life and reduce clearance. The myristoyl (lipid) addition is highly similar to the FDA-approved and blockbuster peptide drug, semaglutide, in which lipidation significantly extended half-life to enable once-a-week dosing. Furthermore, the contribution of PPB to drug success is highly debated, with many FDA-approved drugs showing high PPB. Future detailed PK/PD studies will comprehensively evaluate drug distribution.

Responses to Reviewer #4:

Comment 1:

Reviewer Comment: “The identification of focal adhesions using FAK antibodies is not convincing. Colocalization with another component of focal adhesions, such as paxillin, vinculin, or talin, along with the observed punctate staining would be more convincing. Although his comment applies to both Figure 5d and Supplementary Fig. 9, it is especially true for Supplementary Fig. 9 where red arrows are allegedly pointing to focal adhesions identified using anti-FAK alone. However, these alleged focal adhesions are very difficult, if not impossible, to see, thus making their presence questionable. This issue also makes the quantitation that is provided very difficult to assess.”

RESPONSE: We have added new immunofluorescence data in SK-MEL-147 and WM88 cells with FAK, paxillin, phalloidin (F-actin), and DAPI staining to comprehensively evaluate the effects of 2012 on the focal adhesion (Fig. 5, Supp Fig. 12-14). Focal adhesions can be effectively identified and quantified by FAK/paxillin co-localization at terminating ends of F-actin (phalloidin) staining. Details on the assay and quantitation using the Cytation C10 confocal reader have been described in the Materials & Methods section.

Comment 2:

Reviewer Comment: “Given that patients with melanoma don't die from their primary tumor(s) and that this peptide is being promoted as a potential future therapeutic, it would be more clinically relevant to also examine the mouse lungs or other tissues to determine whether the peptide either prevented metastasis or reduced metastatic tumor growth once metastasis had occurred. In the Methods section, the authors mention that a gross necropsy was performed to examine the presence or absence of metastasis. Were metastases quantitated or were metastatic tissues collected for histology and quantitation? Such data would significantly enhance the importance of this work.”

RESPONSE: We thank the reviewer for this point. Although we agree that melanoma metastasis is a critical feature of human disease progression and mortality, the scope of this initial report on 2012 is focused on primary tumor growth. The B16 model as a primary tumor model (flank injection) is mainly limited to primary tumor growth, as distant metastases are rarely detected in this scenario. We have updated the discussion section to clarify that our data are only applicable to primary tumor growth. Cancer metastasis models will be examined in future publications, including use of genetically engineered mouse models, tail vein injection, and patient derived xenograft models.

Comment 3:

Reviewer Comment: “The authors do not reference the highly relevant Mousson et al. paper (Cancers, 2021, 13, 1871). The authors should consider referencing this paper because it provides further support for their approach.”

RESPONSE: We thank the reviewer for pointing out this publication. A reference has been added for this publication.

Comment 4:

Reviewer Comment: “Myristoylated hydrocarbon-stapled 2012 is far more stable than the native peptide 1907. What about the control peptides 2014 and 2020? Peptide 2020 is used as a control in the in vitro cellular assays and knowing its stability profile would help in evaluating its relevance as a good peptide control. In this same regard, is peptide 2020 cell permeable?”

RESPONSE: Trypsin-protease assays have been added for control peptide 2020, which shows a highly similar stability profile to 2012 (Fig. 5c). Cell permeability tests have been done with TAMRA-2020, which shows that 2020 has >3-fold permeability compared to TAMRA-alkyne control, indicating suitability as a control peptide (Fig. 5b).

Comment 5:

Reviewer Comment: “The functional in vitro studies were performed on SK-MEL-147 melanoma cells. To show broader applicability to other cancers it would be helpful to have these assays performed on another cancer cell line, such as the SK-BR-3 breast cancer cell line the investigators used for other assays in the manuscript.”

RESPONSE: We have added dose-response immunofluorescence data showing dose-dependent delocalization of FAK by peptide 2012 in multiple cancer cell lines, including SK-Mel-147, WM88, and SKBr3. The CTG cell viability assay was also performed in additional cell lines: WM88, ONDA7, NHEM, HEPG2, and LX2 (Fig. 6a-d).

Comment 6:

*Reviewer Comment: “The authors do not describe what measurements were taken to determine tumor size in their in vivo studies nor do they indicate how these measurements were used to calculate tumor volume. A common, but potentially inaccurate, method is to measure the length and width of the tumor with calipers and then assume that the width and depth of the tumor are equal (which can lead to large errors in tumor volume estimates) and that π is equal to 3 (a reasonable approximation). By making these assumptions, a simplified version of the formula for an ellipsoid can be derived, and this formula ($\text{Volume} = \text{length} * \text{width}^2/2$) is often used in the literature but can lead to inaccurate volume estimates as noted above. When all three dimensions of the tumor are measured, the full formula of an ellipsoid can be used to generate more accurate volume estimates. Regardless of what method was used, the authors need to provide this information in their Methods section.”*

RESPONSE: The SC (flank) tumor volume estimation (mm³) was made in accordance with the formula ($a^2 * b/2$) where a is the smallest diameter and b is the largest diameter. The methods section has been updated to report the method of tumor volume calculation.

Comment 7:

Reviewer Comment: “One-way ANOVA was used in several cases when comparing results between multiple groups or treatments, and results of these analyses were provided as a statistically significant p values—meaning that the overall analysis of data from all groups

provided a statistically significant result. However, it does not appear that multiple comparisons tests were performed, and yet asterisks were placed over specific groups, suggesting that these particular groups/treatments were statistically significant when compared to another group or set of groups. Furthermore, in the legend for Supplementary Fig. 9, the authors specifically show/state that their data generated a p value < 0.0001 for both 1 μM 2012 vs DMSO and 1 μM 2036 vs DMSO, implying that they did a multiple comparison test. If so, what multiple comparisons test(s) was/were used to make these assessments? In Supplementary Fig. 9, there appears to be no variation (error bars) in the DMSO control. Does this reflect the actual data or were values in each independent experiment normalized to that experiment's control, thus making all values in the control equal to 1? If the latter explanation is true, a normal one-way ANOVA that includes the control would be inappropriate because all variability in the control would have been eliminated during the normalization process. If so, the authors should consider not normalizing their data or normalizing in a manner that will retain the variability in their control. Otherwise, they may have to log transform their data and use a one-way randomized block ANOVA (same as repeated measures in Prism) with an appropriate multiple comparisons test. Consulting a statistician may be helpful in this case.”

RESPONSE: We thank the reviewer for pointing this out. The Dunnett multiple comparison test was performed for One-way ANOVA analyses in Figure 5, Supplementary Figures 11, 13, 15. These statistical details, including multiple comparison test corrected p-values, have been reported and clarified in the figure legend and methods section. Supplementary Fig. 11 has been corrected to include the variation in the DMSO control group.

Comment 8:

Reviewer Comment: “Two-way ANOVA was performed for the data shown in Figure 5j. Again, a multiple comparison test would need to be performed to determine whether individual time points are significant. What multiple comparison test was performed to determine that data for days 22, 26, and 29 were statistically significant?”

RESPONSE: The Dunnett multiple comparison test was performed for Fig. 5j (now Fig. 6i). The multiplicity adjusted P values were calculated and reported. The family-wise alpha threshold is 0.05 with a 95% confidence interval. These details have been added to the figure legend.

Comment 9:

Reviewer Comment: “The Discussion could be shortened somewhat by eliminating unnecessary reiteration of the approach and results in the second and possibly the third paragraph. A short review of the results to put the discussion in context is valuable, but it seems that there is too much reiteration in these two paragraphs.”

RESPONSE: We thank the reviewer for this constructive suggestion. We have re-focused the discussion section to reflect this comment.

Minor Comments:

Reviewer Comments: “On line 443, it appears the word “described” is missing as in “previously described.” On line 483, “E.coli” should be italicized “E. coli.” On line 485, the first word on the line should be 1-thiogalactopyranoside. In the paragraph encompassing lines 489-497, the three

references to “gluthione” should all be “glutathione.” On line 530, it seems that the word “was” should be “were.” Given that the investigators potentially have a negative control peptide 2020, why was it not used in the *in vivo* studies? Was this a matter of expense? If so, it is understandable, although comparing 2012 to an available similar negative control peptide rather than the vehicle would have been preferred. The authors should consider showing all data points instead of bar graphs or showing a combination of both. This allows readers to get a better sense of the actual data. When comparing panels f and g in Fig. 5, it appears that defactinib reduces cell viability through some mechanism other than apoptosis. Is this the authors' interpretation of the data as well? In panel g (Annexin V assay), defactinib-treated SK-Mel-147 cells exhibit a concentration-dependent decrease in luminescence. Is this because the cells are dead or dying by some mechanism other than apoptosis? Was necrosis also assayed? If so, what was the result? Although defactinib is not the focal point of this study, it is used as a control, and therefore, understanding its mode of action compared to peptide 2012 is relevant. According to the supporting documents, only female mice were evaluated in the reported *in vivo* studies. This should be noted in the “*In vivo melanoma model*” section of the Methods. A short note as to why only female mice were used in the study may also be appropriate.”

RESPONSE: All grammatical errors noted above have been corrected. Bar graphs have also been updated to show all replicate values of experiments. This initial report for novel peptide 2012 is primarily focused on proof-of-concept of pharmacological FAT domain inhibition as a novel strategy for tumor inhibition. Additional *in vivo* studies, including control compounds, dose-response, PDX models, etc., will be performed in future studies. It has been reported that FAK-kinase inhibitors reduce cell number through inhibition of cell cycle progression and not apoptosis (Slack-Davis et al. J Biol Chem. 2007). Based on our Annexin V data, we make a similar conclusion. Necrosis was not examined in this report, however may be investigated in the future. Future differentiation between 2012 and FAK-kinase inhibitors has been clarified in the discussion section. Only female mice were utilized in studies performed by the Experimental Mouse Shared Resource due to less aggressive nature noted by the facility staff. Male vs female sex differences will be explored in future publications.

Responses to Reviewer #5:

Comment 1A:

Reviewer Comment: “NMR studies in Fig.3. The authors run the chemical shift perturbation experiment by employing FAT domain at 100 micromolar concentration and the peptide at the following concentrations: 5,10, 50 micromolar (and lower...). Is it not clear if the “saturation condition” was reached: the authors should run the same HSQC experiment at higher peptide concentrations, let’s say between 75 uM and 100 uM, and demonstrate that the resulting HSQC spectra are identical (=no more changes are occurring) respect to what observed at 50 micromolar. This is a prominent issue as Figure 3 A is not clear: a better larger image might help readers to better understand the data. Moreover, such NMR studies are usually analyzed by reporting in graph (histograms) the chemical shifts and/or intensity deviations of the apo FAT domain (first control HSQC with no peptide and DMSO) respect to completely peptide bound FAT domain (=saturation point) for each single residue over the whole FAT sequence. The authors should produce such graphs, choose a threshold for chemical shifts and/or intensity variations and color consequently the residues most affected by binding on the 3D protein structure. Chemical

shifts variations should be averaged values keeping into accounts $15N$ and $1H$ deviations (Please see for example Morin G, Fradet-Turcotte A, Di Lello P, Bergeron-Labrecque F, Omichinski JG, Archambault J. A conserved amphipathic helix in the N-terminal regulatory region of the papillomavirus E1 helicase is required for efficient viral DNA replication. *J Virol.* 2011 Jun;85(11):5287-300. doi: 10.1128/JVI.01829-10. Epub 2011 Mar 30. PMID: 21450828; PMCID: PMC3094988., Di Lello P, Miller Jenkins LM, Mas C, Langlois C, Malitskaya E, Fradet-Turcotte A, Archambault J, Legault P, Omichinski JG. p53 and TFIIEalpha share a common binding site on the Tfb1/p62 subunit of TFIIF. *Proc Natl Acad Sci U S A.* 2008 Jan 8;105(1):106-11. doi: 10.1073/pnas.0707892105. Epub 2007 Dec 26. PMID: 18160537; PMCID: PMC2224167; Molecular Recognition of Paxillin LD Motifs by the Focal Adhesion Targeting Domain Maria K. Hoellerer, Martin E.M. Noble, Gilles Labesse, Iain D. Campbell, Jörn M. Werner, Stefan T. Arold, <https://doi.org/10.1016/j.str.2003.08.010>.... but there are many NMR related manuscripts showing such kind of analyses).”

RESPONSE: We thank the reviewer for this constructive comment. HSQC experiments were also run at 100 micromolar (the maximum concentration of 1907 that we screened against FAT). Saturation is dependent on the residue. As seen in Supplemental Figure 6, which is a contour plot of specific residues, saturation is achieved at various concentrations. For residues in the 2-3 helices, we see it at as low as 10 micromolar. However, for residues at the 1-4 helices, those residues appear to be saturated at 100 micromolar. Figure 3A has been expanded upon and cleared up. A histogram analyzing the changes in intensity compared to residue number, with prominent CSPs mapped to the 3D structure, has been added to Figure 3 as well.

Comment 1B:

*Reviewer Comment: “Figure 3 panel b is not clear: the peaks should be reported as contour lines and not with solid filling. Panel c: it is very difficult to get quantitative K_D values by NMR especially under strong binding / slow exchange conditions, so what equation did the authors employ to determine K_D values (See also work by Farmer (Farmer BT 2nd, Constantine KL, Goldfarb V, Friedrichs MS, Wittekind M, Yanchunas J Jr. Robertson JG, Mueller L. *Nat. Struct. Biol.* 1996; 3:995–997.))? The employed equation and a proper reference should be introduced and nevertheless, it is again important to know if the last point of titration corresponds to the saturation point to get a reliable k_d estimate. In summary the NMR study appears very qualitative and better / quantitative interpretation of data -as suggested above- is necessary. Graphs of chemical shifts and/or intensity changes vs residue numbers need to be introduced; nevertheless, an overlay of HSQC spectra at two high concentrations of peptides for example 50 μM and 75-100 μM is necessary to demonstrate that saturation conditions were reached at 50 μM peptide concentration. Moreover, the authors compared the HSQC spectrum of protein with 1% DMSO only with HSQC spectra of protein/peptides -with peptide at diverse concentrations-. Does 1% DMSO correspond to the condition 50 micromolar peptide or was it kept constant at all peptide points?”*

RESPONSE: As discussed above, we have added more quantitative interpretation of the NMR data in Figure 3c (histogram). The peaks for all tested concentrations of 1907 as contour lines is reported in Supplemental Figure 6. Information about the equation that was used to calculate K_D values is included in the Methods and Materials section. We are confident that we reach saturation for each residue where a K_D was measured since, as evidenced in Supplemental Figure 6, each

integration does completely disappear at high concentrations of 1907 (e.g., 100 micromolar). 1% DMSO was kept constant across all peptide points, including no peptide.

Comment 2:

Reviewer Comment: "SPR suggests that the peptide 1907 is a dual site binder able to target helices 1-4 and 2-3 in FAK FAT domain. Did the authors try to fit SPR data with a single-site binding model? Is this possible? Why is only a single KD SPR value indicated for the LD2 pep in table 1? It is strange that X-ray can see only electron density at the 2-3 site Could the NMR shifts - observed at site 1-4- be related to conformational movements rather than direct binding? I mean small conformational movements of the protein after binding of the peptide only at the 2-3 site (?). This point can be eventually better addressed with some kind of displacement experiment by employing a ligand specific for the 1-4 single site (if it is available) and an FPA assay. ITC measurements between FAT/1907 could also give additional information on the binding stoichiometry."

RESPONSE: A single-site binding model was initially attempted for peptide 1907, however due to the shape of the SPR sensogram and clear observation of two association and dissociation rates, the two-site binding model provided a more accurate fit. Two separate association/dissociation rates for LD2 were not observed by SPR, likely due to the similar K_D values between helix 2-3 and helix 1-4 sites. Because 1907 demonstrated sub-micromolar binding at only one site, it is not unexpected that electron density was only detected at the single helix 2-3 site. In qualitatively assessing the NMR results, some residues in the 1-4 domain may theoretically undergo CSPs in response to binding in the 2-3 site. However, the residues that undergo such a change are not typically involved in binding with LD2, and we have pointed this out in the main text. We appreciate the suggestion from the reviewer to further elucidate the binding between 1907 and FAT and agree that such experiments are outside the scope of this initial report. Nonetheless, we have shown with SPR, NMR, and X-ray crystallography experiments that 1907 appears to have higher binding affinity to the helix 2-3 site than the helix 1-4 site.

Comment 3:

Reviewer Comment: "CD data, Figure S5 the author say "...demonstrate that hydrocarbon stapling can increase α -helical content by 10 % in aqueous sodium phosphate solution (pH 7.0)". They should clearly indicate how they calculated the 10% increase."

RESPONSE: Secondary structure analysis for CD data, including % alpha helicity, was performed using the CDSSTR method⁴⁶ on data reference set 4 basis spectra.

Comment 4:

Reviewer Comment: "Minor: in the methods 16 scans per "FIP" ...should be "FID"

RESPONSE: This has been corrected.

Responses to Reviewer #1:

Comment 1:

I have only re-read the biological consequences data. I think this manuscript has been improved by the authors. The points I raised have been somewhat addressed and the case is stronger. There is a couple of bits of information I requested that were not referred to in the rebuttal letter I dont think, for example - point 3 - What happens if the stapled peptides are washed out? Do cells recover? I am sure the authors know the answer to this and could perhaps comment. I also think the time-dependent effects may have been part of the first description/basic characterisation of effects, rather than some follow up experiments for next mechanistic studies as mentioned.

RESPONSE: We thank the reviewer for their contributions. Although not reported in this manuscript but will be included in a future manuscript (pulmonary fibrosis focus), we have performed washout studies in TGFb-stimulated lung fibroblasts utilizing peptide 2012 which showed durable biological effects (reduction aSMA stress fibers, reduction of FAK at focal adhesions) up to 96 h post washout, after which at 120h cells recovered to normal levels of aSMA stress fibers/FAK+ focal adhesions. Data are shown below. Future time course studies will be conducted in melanoma cells to determine the timing of specific biological effects following treatment including the time dependency of focal adhesion complex displacement, as we do not believe these studies significantly change the major conclusions from this initial report of 2012. As mentioned in the first response, per Fig. 5 & 6, and Fig. S19, our data suggest in melanoma cells that maximal focal adhesion displacement occurs at 48h whereas most potent induction of apoptosis occurs at 96h.

[REDACTED]

[

Response to Reviewer #2:

Comment 1: *The authors have responded in a satisfactory manner to my comments*

RESPONSE: We thank the reviewer for their contributions to this manuscript.

Response to Reviewer #3:

Comment 1: *Well done, no further comments*

RESPONSE: We thank the reviewer for their contributions to this manuscript.

Response to Reviewer #4:

The authors have addressed the points I raised in my original review, and I only have a few remaining comments and suggestions, all of them relatively minor, except for the first cautionary note.

Comment #1: *As I suggested the authors do in the original review, they have now included the formula used for estimating tumor volume. The one they have chosen to use is the less accurate but commonly used formula for calculating tumor volume, which is derived (see below) from the formula for the volume of an ellipsoid. This simplified formula can lead to large errors in estimated tumor volume if the width diameter (called the smallest diameter in the authors' response) and depth diameters (not measured in this study but assumed to be the same as the width diameter) of the tumor are substantially different. Regardless, this derived formula is commonly employed in tumor studies and the literature is replete with its use. In many cases, using this formula might not change the conclusions of the experiment because all tumor volumes are calculated using the same formula. However, a problem in interpretation could arise if the unmeasured depth diameters of different tumors were substantially different from each other. To better understand this point, the formula for the volume of an ellipsoid and the derivation of the simplified formula are shown below:*

Ellipsoid volume = $\frac{4}{3} \pi \cdot \text{length}/2 \cdot \text{width}/2 \cdot \text{depth}/2$

If one assumes that $\pi = 3$ (a reasonable approximation) and the width and depth of the tumor are the same, this formula simplifies to the formula used in the paper, which is $\text{width}^2 \cdot \text{length}/2$ or $a^2 \cdot b/2$.

One can readily see that if the depth of two tumors is substantially different, but their length and width are the same, the derived formula will lead to the two tumors being estimated as being the same size when, in reality, their volumes could differ by several-fold. Therefore, caution should be exercised when using the simplified formula.

RESPONSE: We thank the reviewer for highlighting the differences between the ellipsoid volume and simplified formulas to estimate tumor volume. We have added a section in the discussion to address the caveat of missing depth measurements but as the reviewer highlighted, given that all tumors were measured using the same method and the common use of the simplified formula in the field, we do not expect using the simplified formula will change the overall conclusions from this study.

Comment #2: *Minor issues and typos that should be addressed:*

a. On line 898, “Dunnet’s” should be “Dunnnett’s”

b. In Supplementary Figures 12 and 14, I think 1.25 μ M should be 12.5 μ M

c. In the Supplementary Data document, the authors should consider making their figure legends consistent with each other and the actual figures by using all lower-case letters (possibly bold font) for panel labels (see Supplementary Figure 15 where uppercase letters were used) and by enclosing the panel labels in parentheses. Other methods can be used but consistency looks good.

d. Point of clarification for a discrepancy. The manuscript (which is the important document) states that Sidak’s multiple comparisons test was used for data shown in Figure 6i while the rebuttal letter (less important) indicates that Dunnnett’s multiple comparison test was used. Please clarify.

RESPONSE: All minor issues and typos have been addressed. For Figure 6i, Šidák’s multiple comparisons test was utilized and apologize for the discrepancy.

Response to Reviewer #5:

I appreciated the authors performed more experiments and put efforts in replying to my comments, however, they still need to revise the manuscript to further clarify the NMR data as not all my queries have been properly addressed.

Comment #1: *I asked if saturation conditions were reached and the authors replied “Saturation is dependent on the residue. As seen in Supplemental Figure 6, which is a contour plot of specific residues, saturation is achieved at various concentrations. For residues in the 2-3 helices, we see it at as low as 10 micromolar. However, for residues at the 1-4 helices, those residues appear to be saturated at 100 micromolar.”*

Well, saturation means that all binding site positions are occupied, and no more changes are observed in the HSQC spectrum of the protein upon further addition of the peptide. So, in theory from what the authors are saying it is still not clear if saturation was reached. Are they sure that if they increase the concentration of peptide, let’s say from 100 micromolar to 200 micromolar, no more changes in the protein HSQC spectrum will occur? Can they show another figure with overlays of HSQCs of protein alone (DMSO control); protein + peptide at 6 micromolar; protein + peptide at 100 micromolar, and protein + peptide at higher concentration -let’s say 200 micromolar-? I really would like to see if the two HSQCs collected at 100 and 200 micromolar peptide are identical (or almost identical) ...this would prove saturation is reached. I’m not surprised that different residues behave differently, this always happens in this case as there are two sites affected by binding, the residues involved the most in peptide binding will likely saturate early. ...but the complete saturation will be reached only after you won’t observe further changes in any peak of

the HSQC spectrum upon further addition of peptide. KD evaluation by NMR is more accurate if saturation conditions are reached. If you are not sure if saturation was reached a comment in the text should be added.

RESPONSE: We thank the reviewer for their contributions and appreciate the reviewer's comment and concern about not reaching full saturation for the entire HSQC spectrum. However, the NMR studies were originally performed to evaluate the peptide binding differences at the helix 2-3 binding site compared to the 1-4 binding site, to corroborate results obtained via X-ray crystallography/SPR. In that regard, the data clearly confirm that there is stronger binding (and therefore saturation at lower concentrations) at residues located at the helix 2-3 site, which is the main conclusion from these NMR studies. Nonetheless, we have added additional data for 200 uM of peptide 1907, including an additional HSQC overlay, as suggested (Suppl Fig. 7a). Based on these data, we do not believe that complete saturation of the entire HSQC spectrum is achieved by 200uM. Due to the dual-site binding of a 13-mer peptide on a relatively small (14 kDa) interconnected 4-helical bundle, the additional CSPs maybe be due to either global conformational changes or incomplete total saturation of the entire HSCQ spectrum. We have edited the main text to mention incomplete saturation of the entire HSQC spectrum calculation and have changed K_D calculation to "estimation" to reflect this potential caveat. However, we confirm that the data do support that 1907 interacts at a higher affinity site at FAT helix 2-3, with a second weaker interaction at the helix 1-4 site.

Comment #2: *Although the authors claim they run experiments at 100 micromolar peptide concentration this condition is never mentioned in the methods or other parts of the main text of the manuscript but just in Fig.S6. Please add the proper conditions in the methods NMR section.*

RESPONSE: The concentrations have been added to the methods & results section.

Comment #3: *Overall, the supplementary Fig. S6 is unclear and further NMR analyses are needed. In detail, please see Fig.3: The authors should also add to panel a in the Figure 3 the overlay with the HSQC spectrum acquired with 100 micromolar peptide; the analyses shown in panels b) and c) need to be repeated for the 100 micromolar concentration (these analyses can be added as a further Supplementary Figure).*

RESPONSE: We thank the reviewer for their comments. We added details to the figure legend of Fig. S6 and included new NMR analyses as suggested by the reviewer (see Fig. 3a & Fig. S7). These new analyses show the peptide 1907 does indeed bind both the helix 2-3 and helix 1-4 site, albeit with the caveat that full saturation of the entire HSQC was not observed, given differences between 100 and 200 uM.

Comment #4: *Caption of Fig.3: "Residues with 1H-15N" replace with "Residues with 1H-15N"; "incubation of FAT at with 6 μM 1907" erase "at"; "Highlighted residues match those highlighted in panel b." change the text to avoid repeating "highlighted" twice*

RESPONSE: These edits were made in the manuscript.

Comment #5: *Figure 3: Panel d) the y axis label is cropped "DMS" please adjust. Overall, the Figure has low resolution. The authors are invited to improve the Figure. This Figure needs to be enlarged as unclear, the text inside the panels is too small.*

RESPONSE: Figure 3d has been adjusted so that the y-axis label is legible. Furthermore, the resolution has been upgraded on this figure.

Comment #6: Regarding K_D evaluation: Keep in mind that in case of slow exchange you cannot really evaluate well a K_D by NMR and you should add a comment on that; please note K_D can be calculated by NMR with certain equations and you should add a reference for the equation you used please see for example : “NMR methods for the determination of protein–ligand dissociation constants *Progress in Nuclear Magnetic Resonance Spectroscopy* 51(2007)219–242; See also *Journal of Biomolecular NMR* (2022) 76:153–163 <https://doi.org/10.1007/s10858-022-00402-3>.

I have already asked clarifications about K_D evaluation and the author replied: “Information about the equation that was used to calculate K_D values is included in the Methods and Materials section.” But in the methods I cannot see any reference to NMR studies for K_D evaluation but just : “The K_d for each residue was calculated using GraphPad Prism using the One Site – Specific Binding Least Squares fit.”

RESPONSE: A statement about the reliability of K_D determination using slow exchange NMR data compared to fast exchange has been added to the results section of the manuscript. We have cited the following manuscript in the methods section (Williamson, M.P. “Using chemical shift perturbation to characterise ligand binding.” *Prog. Nucl. Magn. Reson. Spectosc.* 2013) to estimate K_D using change in intensity (integration) of the peak opposed to shift movement, given the slow exchange features of this 1907:FAT binding interaction. When estimating the K_D from the NMR data, we graphed the concentration of peptide against the change in peak integration compared to DMSO. An equation that is used to estimate the K_D by NMR has been provided in the methods section.

Response to Reviewer #5:

Comment:

The authors replied to all the points that I raised. They improved clarity of Figure 3, they added the additional Supplementary information (See Fig.s S6-S7) and details to the method section, as I asked. I appreciated they inserted a comment about the limits of their KD evaluation by NMR and the fact that, being in slow exchange, and not having reached completely saturation, the reported Kd values by NMR can be considered only "estimates".

In my opinion the manuscript can be accepted for publication. There are minor mistakes ("typos") that I suggest to correct: 1H-15N ...I and 15 should be superscribed and in the methods section after reporting the equation to evaluate kd, when saying " ΔV is one minus the proportion of peak volume" instead of using the word "proportion" -that sounds unusual- I would say "ratio".

Response:

We thank the reviewer for their constructive comments to improve the quality of the manuscript. We have corrected all typos mentioned above.